



# Calibration of a multi-physics ensemble for greenhouse gas atmospheric transport model uncertainty estimation

Liza I. Díaz-Isaac [1,a], Thomas Lauvaux [1], Marc Bocquet [2], Kenneth J. Davis [1]

[1]Department of Meteorology and Atmospheric Science, The Pennsylvania State University, University Park, USA
[2]CEREA, joint laboratory École des Ponts ParisTech and EDF R&D, Université Paris-Est, Champs-sur-Marne, France
[a]now at: Scripps Institution of Oceanography, University of California, San Diego, CA 92093, USA

*Correspondence to*: Liza I. Díaz-Isaac (lzd120@psu.edu)

**Abstract.**

Atmospheric inversions have been used to assess biosphere-atmosphere $CO_2$ surface exchanges at various scales, but
variability among inverse flux estimates remains significant, especially at continental scales. Atmospheric transport errors are one of the main contributors to this variability. To characterize transport errors and their spatio-temporal structures, we present an objective method to generate a calibrated ensemble adjusted with meteorological measurements collected across a region, here the US upper Midwest in midsummer. Using multiple model configurations of the Weather Research and Forecasting (WRF) model, we show that a reduced number of simulations (less than 10 members) reproduces the transport error
characteristics of a 45-member ensemble while minimizing the size of the ensemble. The large ensemble of 45-members was constructed using different physics parameterization (i.e., land surface models (LSMs), planetary boundary layer (PBL) schemes, cumulus parameterizations and microphysics parameterizations) and meteorological initial/boundary conditions. All the different models were coupled to $CO_2$ fluxes and lateral boundary conditions from CarbonTracker to simulate $CO_2$ mole fractions. Meteorological variables critical to inverse flux estimates, PBL wind speed, PBL wind direction and PBL height,
are used to calibrate our ensemble over the region. Two calibration techniques (i.e., simulated annealing and a genetic algorithm) are used for the selection of the optimal ensemble using the flatness of the rank histograms as the main criterion. We also choose model configurations that minimize the systematic errors (i.e. monthly biases) in the ensemble. We evaluate the impact of transport errors on atmospheric $CO_2$ mole fraction to represent up to 40% of the model-data mismatch (fraction of the total variance). We conclude that a carefully-chosen subset of the physics ensemble can represent the errors in the full
ensemble, and that transport ensembles calibrated with relevant meteorological variables provide a promising path forward for improving the treatment of transport errors in atmospheric inverse flux estimates.

## 1 Introduction

Atmospheric inversions are used to assess the exchange of $CO_2$ between the biosphere and the atmosphere (e.g., Gurney et al., 2002; Baker et al., 2006; Peylin et al., 2013). The atmospheric inversion or "top-down" method combines a prior distribution
of surface fluxes with a transport model to simulate $CO_2$ mole fractions and adjust the fluxes to be optimally consistent with





the observations (Enting, 1993). Large differences often exist among inverse flux estimates independent of the spatial scales (e.g., Gurney et al., 2002; Sarmiento et al., 2010; Peylin et al., 2013; Schuh et al., 2013). These posterior flux uncertainties arise from limited atmospheric data density (Gurney et al., 2002), uncertain prior fluxes (Corbin et al., 2010; Gourdji et al., 2010; Huntzinger et al, 2012) and errors in atmospheric transport (Stephens et al., 2007; Gerbig et al., 2008; Pickett-Heaps et

al., 2011; Díaz Isaac et al., 2014; Lauvaux and Davis, 2014).

Atmospheric inversions based on Bayesian inference depend on the prior flux error covariance matrix and the observation error covariance matrix. The prior flux error covariance matrix represents the statistics of the mismatch between the true fluxes and the prior fluxes, but the limited density of flux observation limits our ability to characterize these errors (Hilton et al., 2013). The observation error covariance describes errors of both measurements and the atmospheric transport model. In

atmospheric inversions the model errors tend to be much greater than the measurement errors (e.g. Gerbig et al., 2003; Law et al., 2008). Additionally, atmospheric inversions assume that the atmospheric transport uncertainties are known and are unbiased, therefore the method propagates uncertain and potentially biased atmospheric transport model errors to inverse fluxes limiting their optimality. Unfortunately, rigorous assessments of the transport uncertainties within current atmospheric inversions are limited. Estimation of the atmospheric transport errors and their impact on $CO_2$ fluxes remains a challenge

(Lauvaux et al., 2009).

A limited number of studies are dedicated to quantify the uncertainty in atmospheric transport models and even fewer attempted to translate this information into the impact on the $CO_2$ mixing ratio and inverse fluxes. The atmospheric Tracer Transport Model Intercomparison Project (TransCom) has been dedicated to evaluate the impact of atmospheric transport models in atmospheric inversion systems (e.g., Gurney et al., 2002; Law et al., 2008; Peylin et al., 2013). These experiments have also

shown the importance of the transport model resolution to avoid any misrepresentation of high frequency atmospheric signals (Law et al., 2008). Diaz Isaac et al., (2014) showed how two transport models with two different resolution and physics but using the same surface fluxes can lead to large model-data differences in the atmospheric $CO_2$ mole fractions. These differences would yield significant errors on the inverse fluxes if propagated into the inverse problem. Errors in horizontal wind (Lin and Gerbig, 2005) and in vertical transport (Stephen et al., 2007; Gerbig et al. 2008; Kretschmer et al., 2012) have been shown to

be important contributors to uncertainties in simulated atmospheric $CO_2$. Lin and Gerbig (2005), for example, estimate the impact of horizontal wind error on $CO_2$ mole fractions and conclude that uncertainties in $CO_2$ due to advection errors can be as large as 6ppm. Other studies have shown that errors in the simulation of vertical mixing has a large impact on simulated $CO_2$ and inverse flux estimates (e.g., Denning et al., 1995; Stephens et al., 2007; Gerbig et al., 2008). Therefore, some studies have evaluated the effects that planetary boundary layer height (PBLH) has on $CO_2$ mole fractions (Gerbig et al., 2008;

Williams et al., 2011; Kretschmer et al., 2012). Approximately 3 ppmv uncertainty in $CO_2$ mole fractions have been attributed to PBLH errors over Europe during the summer time (Gerbig et al., 2008; Kretschmer et al., 2012). These studies have attributed the errors to the lack of sophisticated subgrid parameterization, especially PBL schemes and land surface models (LSMs). This led other studies (Kretschmer et al., 2012; Lauvaux and Davis, 2014; Feng et al., 2016) to evaluate the impact of different PBL parameterizations on simulated atmospheric $CO_2$. These studies have found systematic errors of several ppm



in atmospheric $CO_2$ that can generate biased inverse fluxes estimates. While there is an agreement that errors in the vertical mixing and advection schemes can affect directly the inverse fluxes, other components of the model physics (e.g. convection, large-scale forcing) have not been carefully evaluated.

Atmospheric transport models have multiple sources of uncertainty including the boundary conditions, initial conditions,

model physics parameterization schemes and parameter values. With errors inherited from all of these sources, ensembles have become a powerful tool for the quantification of atmospheric transport uncertainties. Different approaches have been evaluated in the carbon cycle community to represent the model uncertainty: (1) the multi-model ensembles that encompass models from different research institutions around the world (e.g. TransCom experiment; Gurney et al., 2002; Baker et al., 2006; Patra et al., 2008; Peylin et al., 2013; Houweling et al., 2010), (2) multi-physics ensembles that involve different model physics

configurations generated by the variation of different parameterization schemes from the model (e.g., Kretschmer et al., 2012; Yver et al., 2013; Lauvaux and Davis 2014; Angevine et al., 2014; Feng et al., 2016; Sarmiento et al, 2017) and (3) multi-analysis (i.e., forcing data) that consists of running a model over the same period using different analysis fields (where perturbations can be added) (e.g., Lauvaux et al., 2009; Miller et al., 2015; Angevine et al., 2014). These ensembles are informative (e.g., Peylin et al., 2013; Kretschmer et al., 2012; Lauvaux and Davis 2014), but have some shortcomings. In some

cases, the ensemble spread includes a mixture of transport model uncertainties and other errors such as the variation in prior fluxes or the observations used. Other studies have only varied the PBL scheme parameterizations. None of these studies have carefully assessed whether or not their ensemble spreads represent the actual transport errors.

In the last two decades, the development of ensemble methods has improved the representation of transport uncertainty using the statistics of large ensembles to characterize the statistical spread of atmospheric forecasts (e.g. Evensen, 1994a, 1994b).

Single-physics ensemble-based statistics are highly susceptible to model error, leading to under-dispersive ensembles (e.g. Lee et al., 2012a). Large ensembles (>50 members) remain computationally expensive and ill-adapted to assimilation over longer time scales such as multi-year inversions of long-lived species (e.g. $CO_2$). Smaller-size ensembles would be ideal, but most initial-condition-only perturbation methods produce unreliable and overconfident representation of the atmospheric state (Buizza et al. 2005). An ensemble used to explore and quantify atmospheric transport uncertainties requires a significant

number of members to avoid sampling noise and the lack of dispersion of the ensemble members (Houtekamer and Mitchell, 2001). However, large ensembles are computationally expensive. Limitations in computational resources lead to restrictions including the setup of the model (e.g., model resolution, nesting options, duration of the simulation) and the number of ensemble members. It is desirable to generate an ensemble that is capable of representing the transport errors, and that does not include any redundant members.

Various post-processing techniques can be used to calibrate or "down-select" from a transport ensemble of 50 or more members to a subset of ensemble members that represent the model transport errors (e.g., Alhamed et al., 2002; Garaud and Mallet, 2011; Lee et al., 2012a; 2016). Some of these techniques are principal component analysis (e.g., Lee et al., 2012a), K-means cluster analysis (e.g., Lee et al., 2012b) and hierarchical cluster analysis (e.g., Alhamed et al., 2002; Yussouf et al., 2004; Johnson et al., 2011; Lee et al., 2012b; 2016). Riccio et al. (2012), applied the concept of "uncorrelation" to reduce the number



of members without using any observations. Solazzo and Galmarini (2014) reduced the number of members by finding a subset of members that maximize a statistical performance skill such as the correlation coefficient, the root-mean-square error or the fractional bias. Other techniques applied less commonly to the calibration of the ensembles include simulated annealing and genetic algorithms (e.g. Garaud and Mallet, 2011). All these techniques are capable of eliminating those members that are

redundant, and generating an ensemble with a smaller number of members that represents the uncertainty of the atmospheric transport model more faithfully than the larger ensemble.

In this study we start with a large multi-physics/multi-analysis ensemble of 45-members presented in Díaz-Isaac et al. (2018) and apply a calibration process similar to the one explained in Garaud and Mallet (2011). Two principal features characterize an ensemble: reliability and resolution. The reliability is the probability that a simulation has of matching the frequency of an

observed event. The resolution is the ability of the system to predict a specific event. Both features are needed in order to represent model errors accurately. Our main goal is to generate an ensemble that will represent the uncertainty of the transport model with respect to meteorological variables of most importance in simulating atmospheric $CO_2$. These variables are the horizontal mean PBL wind speed and wind direction, and the vertical mixing of surface fluxes, i.e. PBLH. We focus on the criterion that will measure the reliability of the ensemble, i.e. the probability of the ensemble in representing the frequency of

events (i.e. the spatio-temporal variability of the atmospheric state). For the selection of the ensemble, we will use two different techniques, simulated annealing and a genetic algorithm. In a final step, the ensemble with the optimal reliability will be selected by minimizing the biases in the ensemble mean. We will evaluate which physical parameterizations play important roles in balancing the ensembles and evaluate how well a pure physics ensemble can represent transport uncertainty.

## 2 Methods

### 2.1 Generation of the ensemble

We generate an ensemble using the Weather Research and Forecasting (WRF) model version 3.5.1 (Skamarock et al., 2008), including the chemistry module modified in this study for $CO_2$ (WRF-ChemCO$_2$). The ensemble consists of 45-members that were generated by varying the different physics parameterization and meteorological data. The land surface models, surface layers, planetary boundary layer schemes, cumulus schemes, microphysics schemes, and meteorological data (i.e., initial and

boundary conditions) are alternated in the ensemble (see Table 1). All the simulations use the same radiation schemes, both long and shortwave.

The different simulations were run using the one-way nesting method, with two nested domains (Figure 1). The coarse domain (d01) uses a horizontal grid spacing of 30km and covers most of the United States and part of Canada. The inner domain (d02) uses a 10km grid spacing, is centered in Iowa and covers the Midwest region of the United States. The vertical resolution of

the model is described with 59 vertical levels, with 40 of them within the first 2km of the atmosphere. This work focuses on the simulation with higher resolution, therefore only the 10-km domain will be analyzed.





The $CO_2$ fluxes for summer 2008 were obtained from NOAA Global Monitoring Division's CarbonTracker version 2009 (CT2009) data assimilation system (Peters et al., 2007; with updates documented at https://www.esrl.noaa.gov/gmd/ccgg/carbontracker/). The different surface fluxes from CT2009 that we propagate into the WRF-ChemCO$_2$ model are fossil fuel burning, terrestrial biosphere exchange, and exchange with oceans. The $CO_2$ lateral

boundary conditions were obtained from CT2009 mole fractions. The $CO_2$ fluxes and boundary conditions are identical for all ensemble members.

### 2.2 Dataset and data selection

Our interest is to calibrate the ensemble over the Midwest U.S. using the meteorological observations available over this region. The calibration of the ensemble will be done only within the inner domain. To perform the calibration, we used balloon

soundings collected over the Midwest region (Figure 1). Meteorological data were obtained from the University of Wyoming's online data archive (http://weather.uwyo.edu/upperair/sounding.html) for 14 rawinsonde stations over the U.S. Midwest region (Figure 1). To evaluate how the new calibrated ensemble impacts $CO_2$ mole fractions we will use in-situ atmospheric $CO_2$ mole fraction data provided by seven communication towers (Figure 1). Five of these towers were part of a Penn State experimental network, deployed from 2007 to 2009 (Richardson et al., 2012; Miles et al., 2012, 2013;

http://dx.doi.org/10.3334/ORNLDAAC/1202). The other two towers (Park Falls-WLEF and West Branch-WBI) are part of the Earth System Research Laboratory/Global Monitoring Division (ESRL/GMD) tall tower network (Andrews et al., 2014), managed by NOAA. Each of these towers sampled air at multiple heights, ranging from 11 m AGL to 396 m AGL.

The ensemble will be calibrated for three different meteorological variables: PBL wind speed, PBL wind direction and planetary boundary layer height (PBLH). We will calibrate the ensemble with the late afternoon data (i.e., 0000 UTC) from

the different rawinsondes. In this study, we use only daytime data, because we want to calibrate and evaluate the ensemble under the same well mixed conditions that are used to perform atmospheric inversions. For each rawinsonde site we will use wind speed and wind direction observations from approximately 300 m above ground level (AGL). We choose this observational level because we want the observations to lie within the well mixed layer, the layer into which surface fluxes are distributed, and the same air mass that is sampled and simulated for inversions based on tower $CO_2$ measurements.

The PBLH was estimated using the virtual potential temperature gradient ($\nabla \theta v$). The method identifies the PBLH as the first point above the atmospheric surface layer where (1) $\nabla \theta v$ is greater than or equal to 0.2 K/km, and (2) the difference between the surface and the threshold level virtual potential temperature is greater than or equal to 3 K ($\theta_{vs} - \theta_v \geq 3K$).

WRF derives an estimated PBLH for each simulation, however the technique used to estimate the PBLH varies according to the PBL scheme used to run the simulation. For example, the YSU PBL schemes estimates PBLH using the Bulk Richardson

number, MYJ PBL scheme uses the TKE to estimate the PBLH and MYNN PBL scheme uses QKE to estimate the PBLH. To avoid any errors from the technique used to estimate the PBLH, we decided to estimate the PBLH from the model using the same method used for the observations. Simulated PBLH will be analyzed at the same time as the observations, 0000 UTC, i.e., late afternoon in the study region.





We analyzed $CO_2$ mole fractions collected from the sampling levels at or above 100m AGL, which is the highest observation level across the MCI network (Miles et al., 2012). This ensures that the observed mole fractions reflect regional $CO_2$ fluxes and not near-surface gradients of $CO_2$ in the atmospheric surface layer (ASL) or local $CO_2$ fluxes (Wang et al., 2007). Both observed and simulated $CO_2$ mole fractions are averaged from 1800 to 2200 UTC (12:00-16:00 LST), when the daytime period

of the boundary layer should be convective and the $CO_2$ profile well mixed (e.g., Davis et al., 2003; Stull, 1988). This averaged mole fraction will be referred to hereafter as daily daytime average (DDA).

**2.3 Criteria**

In this research we want to test the performance of the transport ensemble and try to achieve a better representation of transport uncertainties, if possible using an ensemble with a smaller number of members. A series of statistical metrics are used as

criteria to measure the representation of uncertainty by the ensemble for the period of June 18 to July 21 of 2008. The criteria used for our down-selection process include ranks histograms, rank histogram scores and ensemble bias.

**2.3.1 Talagrand diagram (or rank histogram) and rank histogram score**

The rank histogram and the rank histogram scores are tools used to measure the spread, and hence the reliability of the ensemble (see Figure A1 in Appendix).. The rank histogram (Anderson 1996; Hamill and Colucci 1997; Talagrand et al., 1999) is

computed by sorting the corresponding modelled variable of the ensemble in increasing order and then a rank among the sorted predicted variable from lowest to highest is given to the observation. The ensemble members are sorted to define "bins" of the modelled variable, if the ensemble contains N members, then there will be N+1 bins. If the rank is zero then the observed variable value is lower than all the modelled variable values, and if it is N+1 then the observation is greater than all of the modelled values. If the ensemble is perfectly reliable, the rank histogram should be flat (i.e. flatness equal to 1). This happens

when the probability of occurrence of the observation within each bin is equal. A rank histogram that deviates from the flat shape implies a biased, overdispersive or underdispersive ensemble. A "U-shaped" rank histogram indicates that the ensemble is underdispersive, normally in this type of ensemble the observations tend to fall outside of the envelope of the ensemble and indicates a lack of variability. A "central-dome" (or "A-shaped") histogram indicates that the ensemble is overdispersive; this kind of ensemble has an excess of variability. If the rank histogram is overpopulated at either of the ends of the diagram, then

this indicates that the ensemble is biased.

The rank histogram score is used to measure the deviation from flatness of a rank histogram:

$$\delta = \frac{N+1}{NM} \sum_{j=0}^{N} (r_j - \bar{r})^2 \ , \tag{1}$$

and should ideally be close to 1 (Talagrand et al., 1999; Candille and Talagrand, 2005). In Eq.(1), $N$ is the number of models,

$M$ is the number of observations, $r_j$ the number of observations of rank j, and $\bar{r} = M/(N+1)$ is the expectation of $r_j$. In theory, the optimal ensemble has a score of one (1) when enough members are available. A score lower than one would indicate



overconfidence in the results, with an ensemble matching the observed variability better than statistically expected. Having a score smaller than one would not affect the selection process. Nevertheless, a flat rank histogram does not necessarily mean that the ensemble is reliable or has enough spread. For example, a flat histogram can still be generated from ensembles with different conditional biases (Hamill, 2001). The flat rank histogram can also be produced when covariances between samples

are incorrectly represented. Therefore, additional verification analysis has to be introduced to certify that the calibrated ensemble has enough spread and is reliable. We introduce hereafter several additional metrics used to evaluate the ensemble.

**2.3.2 Ensemble bias**

Atmospheric inverse flux estimates are highly sensitive to biases. The bias, or the mean of the model-data mismatches, was used to assist the selection of the calibrated sub-ensemble. We identify a sub-ensemble that has minimal bias,

$$Bias = \frac{1}{M}\sum_{i=1}^{M}(p_i),$$ (2)

where $p_i$ is the difference between the modeled wind speed, direction or PBLH, and the observed value, $M$ is the number of measurements and $i$ sums over each of the rawinsonde measurements.

**2.4 Verification methods**

Different statistical tools were used to evaluate both the large ensemble (45-member) and calibrated ensemble, these statistics include Taylor diagrams, spread-skill relationship, and ensemble root mean square deviation (RMSD). These statistical analyses will be used to describe each member performance (standard deviations and correlations), ensemble spread (root mean square deviation) and error structures in space (error covariance), these are some of the important aspects for the ensemble evaluation.

We use Taylor diagrams to describe the performance of each of the models of the large ensemble (Taylor, 2001). The Taylor diagram relies on three nondimensional statistics: the ratio of the variance (model variance normalized by the observed variance), the correlation coefficient, and the normalized center root-mean square (CRMS) difference (Taylor, 2001). The ratio of the variance or normalized standard deviation indicates the difference in amplitude between the model and the observation. The correlation coefficient measures the similarity in the temporal variation between the model and the observation. The

CRMS is normalized by the observed standard deviation and quantifies the ratio of the amplitude of the variations between the model and the observations.

To verify that the ensemble captures the variability in the model performances across space and time, we computed the relationship between the spread of the ensemble and the skill of the ensemble over the entire data set (i.e. spread-skill relationship). The linear fit between the two parameters measures the correlation between the ensemble spread and the

ensemble mean error or skill (Whitaker and Lough, 1998). The ensemble spread is calculated by computing the standard deviation of the ensemble and the mean error by computing the absolute difference between the ensemble mean and the observations. Ideally, as the ensemble skill improves (the mean error gets smaller), the ensemble spread becomes smaller, and





vice versa. Compared to the rank histograms, spread-skill diagrams represent the ability of the ensemble to represent the errors in time and space.

The spread of the ensemble is evaluated in time, using the Root Mean Square Deviation (RMSD). The RMSD does not consider the observations as we take the square root of the average difference between model configuration and the ensemble mean.

Additionally, we use the mean and standard deviation of the error (model-data mismatch) to evaluate the performance of each of the member selected for the calibrated ensembles.

Transport model errors in atmospheric inversions are described in the observation error covariance matrix, hence in $CO_2$ mole fractions ($ppmv^2$). Therefore, we evaluate the impact of the calibration on the variances of $CO_2$ mole fractions. For the covariances, we compare the spatial extent of error structures between the full ensemble and the reduced-size ensembles by

looking at spatial covariances from our measurement locations. The limited number of members is likely to introduce sampling noise in the diagnosed error covariances. We also know that the full ensemble is not a perfect reference, but we believe is less noisy. The covariances were directly derived from the different ensembles to estimate the increase in sampling noise as a function of the ensemble size.

## 2.5 Calibration methods

In this study, we want to test the ability to reduce the ensemble from 45-members to an ensemble with a smaller number of members that is still capable of representing the transport errors and does not include members with redundant information. We use the Garaud and Mallet (2011) technique to define the size of the calibrated sub-ensemble that each optimization technique will generate, the size of the sub-ensemble was determined by dividing the total number of observations by the maximum frequency in the rank histogram. We are going to generate sub-ensembles of three different sizes (number of

members) to evaluate the impact that an ensemble size has on the representation of atmospheric transport uncertainties. Each of the ensembles will be calibrated for the period of June 18 to July 21 of 2008.

Two optimization methods, simulated annealing (SA) and a genetic algorithm (GA), are used to select a sub-ensemble that minimizes the rank histogram score (δ), which is the criterion that each algorithm will use to test the reliability of the ensemble. Each method will select a sub-ensemble that best represents the model uncertainties of PBL wind speed, PBL wind direction

and PBLH.

In this study, SA and GA techniques will randomly search for the different combinations of members and compute the flatness score. Both techniques generate a sub-ensemble (S) of size N. For the first test, we will use these algorithms to choose the combination of members that optimize the score of the reduced ensemble $J(S)$ (i.e., rank histogram score (δ as defined in Eq.(1)) for each variable. With this evaluation, we determine if each optimization technique yields similar calibrated

ensembles, and if the calibrated ensembles are similar among the different meteorological variables. In the second test, we calibrate the ensemble for all three variables simultaneously, where we use the sum of the score squared: $[J(S)]^2$ (i.e. $\delta^2$ ):

$$[J(S)]^2 = [J_{wspd}(S)]^2 + [J_{wdir}(S)]^2 + [J_{pblh}(S)]^2, \tag{3}$$




to control acceptance of the sub-ensembles. In Eq. (3), $J_{wspd}(S)$, $J_{wdir}(S)$ and $J_{pblh}(S)$ are the scores of the sub-ensemble for PBL wind speed, PBL wind direction and PBLH respectively.

### 2.5.1 Simulated Annealing

Simulated annealing (SA) is a general probabilistic local search algorithm, described by Kirkpatrick et al. (1983) and Cerny et

al. (1985) as an optimization method inspired from the process of annealing in metal work. Based on the Monte-Carlo iteration solving method, SA finds the global minimum using a cost function that gives to the algorithm the ability to jump or pass multiple local minima (see Figure A2 in Appendix). In this case the optimal solution is a sub-ensemble with a rank histogram score close to 1.

The SA starts with a randomly selected sub-ensemble. The current state (i.e, initial random sub-ensemble) has a lot of

neighbours states (i.e., other randomly generated sub-ensembles) in which a unit (i.e., model) is changed, removed or replaced. Let $S$ be the current sub-ensemble and $S'$ be the neighbor sub-ensemble. $S'$ is a new sub-ensemble (i.e., neighbor) that is randomly built from the current sub-ensemble with one model added, removed or *replaced*. To minimize the score $J$, only two transitions to the neighbours are possible. First transition, if the score of the neighbour sub-ensembles $J(S')$ is lower than the current sub-ensemble $J(S)$, then $S'$ becomes the current sub-ensemble and a new neighbour sub-ensemble is generated. Second

transition, if the score of the neighbour sub-ensemble $J(S')$ is greater than the current sub-ensemble J(S), moving to the neighbour S' only occurs through an acceptance probability. This acceptance probability is equal to $exp\left(-\frac{J(S')-J(S)}{T}\right)$ and it only allows the movement to the neighbor S' if $u < exp\left(-\frac{J(S')-J(S)}{T}\right)$. For the acceptance probability, $u$ is a random number uniformly drawn from [0,1] and T is called temperature and it decreases after each iteration following a prescribed schedule. The acceptance probability is high at the beginning and the probability of switching to neighbour less at the end of the

algorithm. The possibility to select a less optimal state $S'$, i.e., with higher $J(S')$ is meant to escape local minima where the algorithm could remain trapped.

When the algorithm reaches the predefined number of iterations, we collect only the accepted sub-ensemble $S$ and their respective scores $J(S)$. When the algorithm finishes with the iterations, we choose the ensemble that has both the smallest rank histogram score and lowest bias among the different sub-ensembles (see Section 2.7). The number of iterations was defined

by sensitivity test and repetitively of the experiments (see Section 2.6)

### 2.5.2 Genetic Algorithm

A genetic algorithm (GA) is a stochastic optimization method that mimics the process of biological evolution, with the selection, crossover and mutation of a population (Fraser and Burnell, 1970; Crosby, 1973; Holland, 1975). Let $S_i$ be an individual; that is, a sub-ensemble, and let $P = \{S_1, \dots, S_i, \dots, S_{N_{pop}}\}$ be a population of $N_{pop}$ individuals (see Figure A3 in

appendix). As a first step in the GA a random population is generated (denoted $P^0$). Then this population will go through two steps (1) selection and (2) crossover. In the selection step, we select half of the best individuals with respect to the score (i.e.,



summation of the score of three variables $J(S)$). For the second step, a crossover among the selected individuals occurs when two parents create two new children by exchanging some ensemble members. A new population is generated with $N_{pop}/2$ parents and $N_{pop}/2$ children.

This process is repeated until it reaches the specified number of iterations. This algorithm will provide at the end a population

of individuals with a better rank histogram score than the initial population. Out of all those individuals we choose the sub-ensemble with the best score for the three variables (i.e., wind speed, wind direction and PBLH) and with a smaller bias than the large ensemble.

**2.6 Parameterization of the selection algorithms**

Various inputs are required to guide the selection algorithms. For example, we typically need to choose the initial and final

temperature ($T_0$ and $T_f$) for the SA and its schedule, the best population size ($N_{pop}$) for the GA and the number of iterations for each algorithm. The temperature of the SA, the $N_{pop}$ of the GA and the number of iterations were chosen by running the algorithms multiple times and confirming that the system reached similar solutions with independent minimization runs. If similar solutions were not achieved within multiple SA or GA runs, the algorithm parameters were altered to increase the breadth of the search. For the SA we found that 20,000 iterations yielded similar solutions after multiple runs of the algorithm.

For the GA, 30 to 50 iterations were sufficient as long as the ensemble was smaller than 8-members. For an ensemble of 10-members we needed to increase to 100 iterations. Another factor that was important in the SA was the initial temperature used in the algorithm and the temperature decrease for each iteration. While the temperature is high, the algorithm will accept with more frequency the poorer solutions; as the temperature is reduced, the acceptance of poorer solutions is reduced. Therefore, we needed to provide an initial ($T_0$) and final ($T_f$) temperature that allowed the system to reduce its acceptance condition

gradually and to search more combinations of members to identify the best solution or sub-ensemble. We determine the optimal parameters for SA by the maximum number of ensemble solutions which indicates that the algorithm explored the largest space of solution with $T_0$ equal to 20 and $T_f$ equal to 1e-3. For GA the larger the population, the more we can explore the space to find an optimal solution. We found that a $N_{pop}$ of 280 individuals was the value that produced similar solutions (sub-ensembles) after multiple runs.

**2.7 Selection of the optimal reduced sized-ensembles**

The selection process is performed in three distinct steps to ensure that the final calibrated ensembles will be the optimal combinations of model configurations (Figure 2). First, the flatness of the rank histograms will control the acceptance of the calibrated sub-ensembles by the selection algorithms (see Figure A1 in Appendix). The flatness is defined by equation (1) for the single-variable calibration and equation (3) for the calibration of the three variables simultaneously. The algorithm selects

multiple sub-ensembles with a rank histogram score smaller than six for each individual meteorological variable, or smaller than the original ensemble score if higher than six (see Figure 2 and Table 2). In general, the lowest scores are found for PBLH



and the highest for wind speed, as shown in Figure 3. As a second step, sub-ensembles accepted by SA and GA algorithms with a bias larger than the bias of the full ensemble are filtered out. This step is critical to avoid the selection of biased ensembles as discussed by Hamill et al. (2001). Finally, the remaining calibrated ensembles are compared among SA and GA techniques to identify if both algorithms provide a common solution. If multiple common solutions were identified, the final

sub-ensemble was determined by the solution with the smallest score and bias. However, if no common solution was found by both techniques, the final sub-ensemble corresponds to the smallest score among the different solutions that share >50% of the same model configurations.

### 3. Results

### 3.1 Evaluation of the large ensemble

In this section, we evaluate the performance of the large ensemble. Our goal is to test the ensemble skill (ability of the models to match the observations) and the spread (variability across model simulations to represent the uncertainty). We will evaluate the skill and the spread for PBLH, PBL wind speed, and PBL wind direction across the region of study using afternoon (0000 UTC) rawinsonde observations.

### 3.1.2 Model skill

We evaluate the performance of the different models of the 45-member ensemble by computing the normalized standard deviation, normalized center root mean square and correlation coefficient for wind speed (Figure 4a), wind direction (Figure 4b) and PBLH (Figure 4c) (Taylor, 2001). The majority of the model configurations produce winds speeds and directions with higher standard deviations (more variability) than the observations, whereas the simulations over- and under-estimate PBLH variability depending on the model configuration. The model-data correlations with wind speed and wind direction are between

0.4 and 0.7, whereas the PBLH shows a smaller correlation, between 0.3 and 0.6. The range of modeled PBL heights will provide a wide spectrum of alternatives to select the optimal calibrated sub-ensemble. However, wind speed and wind direction do not show much difference among the different models. This limited spread potentially reduces the selection of the model configurations to produce a sub-ensemble that matches the observed variability.

### 3.1.3 Reliability and spread of the ensemble

We illustrate the ensemble spread and how well this ensemble encompasses the observations using the time series of the simulated and observed meteorological variables. Figure 5 shows the time series of the ensemble spread for wind speed, wind direction and PBLH at the GRB (Figure 5 a,c,e) and TOP (Figure 5 b,d,f) sites. The time series show qualitatively that simulated wind speed (Figure 5 a-b) and wind direction (Figure 5 c-d) have a smaller spread compared to PBLH (Figure 5 e-f). Figure 5 shows how the ensemble can have a small spread and still encompass the observations (i.e., DOY 183 Figure 5c);





and have a large spread and not encompass the observation (i.e., DOY 174 Figure 5e). These time series suggest that the ensemble may struggle to encompass the observed wind speed and wind direction more than the PBLH.

Figure 6 shows the rank histograms of the 45-member ensemble for each of the meteorological variables that we use to calibrate the ensemble (i.e., wind speed, wind direction and PBLH). In these rank histograms we include all 14 rawinsonde sites. All

the rank histograms have a U-shape. U-shaped histograms mean the ensemble is under-dispersive, that is, the model members are too often all greater than or less than the observed atmospheric values (e.g. DOY 178-181, Fig 5b). Each rank histogram has the first rank as the highest frequency, indicating that observations are most frequently below the envelope of the ensemble (e.g. DOY 178-180, Fig 5b). The rank histogram score for each of the variables is greater than one, confirming that we do not have optimal spread in our ensemble. Table 2 shows that both wind speed and wind direction have a higher rank histogram

score (i.e., $\geq 6$) than the PBLH that has a score of 3.2. The ensemble mean wind speed and PBLH shows a small positive bias relative to the observations, averaged across the region, whereas wind direction has a very small negative bias.

Figure 7 shows the spread-skill relationship, another method that we use to examine the representation of errors of the ensemble. Wind direction (Figure 7b) shows a higher correlation between the spread and the skill compared to the PBLH (Figure 7c) and the wind speed (Figure 7a). Therefore, the ensemble has a wider spread when the model-data differences are

larger. The PBLH and wind speed show consistently poorer skill (a large mean absolute error) compared to their spread. This supports the conclusion that the large ensemble is under-dispersive for these variables. None of these variables shows a correlation equal to one; this implies that our ensemble spread does not match exactly the atmospheric transport errors on a day-to-day basis. This feature is common among ensemble prediction systems (Wilks et al., 2011) and should not impair the ability to identify the optimal reduced-size ensembles.

**3.2 Calibrated ensemble**

In this section, we show the results of the calibrated ensembles generated with both SA and GA. Each calibration was performed for three different sub-ensemble sizes; the size of the ensembles is determined using the technique explained in Section 2.4. To compute the size of the sub-ensemble we use the maximum frequency of the rank histogram, in this case the maximum frequency is the left bar ($r_0$) of every rank histogram. This technique yields the result that the calibrated ensemble

should have about 8 to 10 members depending in the variable to be used. Therefore, for this study we will generate 10, 8 and 5-member ensembles using the two calibration techniques.

**3.2.1 Individual variable calibration**

Table 3 shows that both techniques (i.e., SA and GA) were able to find similar combinations of model configurations (i.e., an ensemble that shares more than half of the members) when each meteorological variable was used separately. The

configurations chosen for each sub-ensemble vary significantly across the different variables, with the exception of the 10-member ensemble calibrated using wind speed and wind direction. The majority of the ensembles include model configuration 14. This model configuration, as shown in Díaz-Isaac et al. (2018), introduces large errors for both wind speed and wind



direction, and is selected to allow for sufficient spread of these variables in the sub-ensembles. The final scores of the calibrated ensembles for each variable show that finding a calibrated sub-ensemble that reaches a score of one is not possible for wind speed and wind direction. A sub-ensemble with a score less than or equal to one can be found for PBLH. Figure 8 shows the rank histograms of the different calibrated ensembles (i.e., 10, 8 and 5-member) for each meteorological variables shown in

Table 3. The calibrated ensembles of PBLH (Figure 8 c, f, i) are nearly flat for all ensemble sizes, whereas the 10- and 8-member sub-ensembles keep a slight U-shape for wind speed and wind direction, but are significantly flatter than the original ensemble. The ratio between the expected ($\bar{r}$) and observed frequency of the end members is reduced from 5 (original expected frequency of 0.02 with 0.1 frequency observed) to less than 2 (calibrated expected frequency of 0.1 with 0.15 frequency observed). The smallest rank-histogram score for wind speed and wind direction are obtained with a 5-member

ensemble (Figure 8 g-h).  The biases for all sub-ensembles (Table 3) are similar to or less than the bias of the large ensemble (Table 2).

### 3.2.2 Multiple variable calibration

Table 4 shows the sub-ensembles selected by SA. Each of the sub-ensembles have two simulations in common (i.e., 17 and 33), implying that these models are crucial to build an ensemble that best represents the transport errors for the three variables.

Figure 9 shows the rank histograms of the sub-ensembles shown in Table 4. These rank histograms show that we were able to flatten the histogram relative to the 45-member ensemble for all three meteorological variables. Similar to the individual variable calibration, the rank histogram for wind speed (Figure 9a, d) and wind direction (Figure 9b, e) still show a U-shape which is minimized for the smallest (i.e., 5-member) sub-ensemble (Figure 9g-h). The rank histograms are flatter for the PBLH (Figure 9c, f, i) and the histogram score is closer to one (Table 4) compared to wind speed and wind direction. The rank

histogram scores for all variables are greater than those for one-variable optimization (see Table 4). In addition, all these calibrated sub-ensembles have biases smaller in magnitude than the 45-member ensemble. Both wind speed and PBLH retain an overall positive bias, and wind direction a negative bias. The standard deviations of these three calibrated ensembles are larger than those of the large ensemble, consistent with the effort to increase the ensemble spread.

Using SA and GA techniques and the selection criteria detailed in Section 2.7 (i.e. low mean error of the entire ensemble), we

defined an optimal 5-member sub-ensemble (the optimal solution using both techniques) and nearly identical combinations of members for 10-and 8-member sub-ensembles, with only two model configurations not being shared by both algorithms.  We also find that configuration 14 remains important for the multi-variable calibrated ensembles, as it was for the single-variable calibrated ensembles.

### 3.2.3 Evaluation of the multiple variable calibrated ensemble

Both optimization techniques were able to generate sub-ensembles that reduce the U-shape of the rank histograms while significantly decreasing the number of members in the ensemble. A flatter histogram indicates that the ensemble is more reliable (unbiased) and has a more appropriate (greater) spread. The correlation between spread and skill for the wind direction





increased while wind speed and PBLH remain similar. Therefore, we conclude that the calibrated sub-ensembles are equivalent or even better than the full ensemble to represent the daily model errors.

Figure 10 shows the time series of the different calibrated ensembles generated by the SA algorithm at TOP site. In general there are no major differences among 5- (Figure 10a,d,g), 8- (Figure 10,e,h) and 10-member (Figure 10c,f,i) ensembles. Figure 12 shows how the calibration can increase the spread of the ensemble to the extent of encompassing the observations (e.g., DOY 179 Figure 10 b-c) compared to the full ensemble (Figure 5b). The ensemble spread was reduced after calibration at a few specific points in space and time.

Insight into the physics parameterizations can be gained by evaluating the calibrated ensembles. The LSM, PBL, CP, and MP scheme, and reanalysis choice varies across all of the sub-ensemble members; no single parameterization is retained for all members in any of these categories. However, we also find that the calibrated ensembles rely upon certain physics parameterizations more than others. Figure 11 shows that most of the simulations in the calibrated ensemble use the RUC and Thermal Diffusion (T-D) LSMs in preference to the Noah LSM. In addition, more simulations use the MYJ PBL scheme than the other PBL schemes. The physics parameterizations shown with a higher percentage in Figure 11 appear to contribute more to the spread of the ensemble than the other parameterizations.

We next explore the characteristics of the individual ensemble members that are retained in an effort to understand what member characteristics are important to increase the spread of the ensemble. Figure 12 shows the mean and standard deviation of the residuals for each simulation included in the 5-member ensemble of SA and GA. Ensembles appear to need at least one member with a larger standard deviation to improve the spread for wind speed and wind directions (see member 23 from Figure 12a-b). Additionally, a member that has a large PBLH bias (see member 16 from Figure 12c) appears to be selected, highlighting the need for end members among the model configurations in order to reproduce the observed variability in PBLH. We note here that the model configuration 14 was not selected when calibrating three variables together.

### 3.3 Propagation of transport errors into $CO_2$ concentrations

The calibrated ensembles found in this study were chosen based on the meteorological variables and not on the $CO_2$ mole fractions to avoid the propagation of $CO_2$ flux biases into the solution. We can now propagate these errors, represented by the ensemble spread, into the $CO_2$ concentration space. This straightforward calculation is possible because every model simulation uses identical $CO_2$ fluxes. We present here the transport errors in both time and space with the spread in $CO_2$ mole fractions, comparing the initial (un-calibrated) 45-member ensemble to the calibrated sub-ensembles.

### 3.3.1 $CO_2$ error variances

Figure 13 shows the spread of daily daytime average $CO_2$ mole fractions across the different sub-ensemble sizes at Mead (Figure 13a,d,g,j), West Branch (Figure 13b,e,h,k) and WLEF (Figure 13c,f,i,l). The spread of the DDA $CO_2$ mole fractions of the large ensemble (Figure 13a-c) does not appear to differ in a systematic fashion from the spread of the calibrated small-size ensembles (Figure 13 d-l). While the calibration has increased the average ensemble spread, none of the ensembles



consistently encompasses the observations, either in terms of meteorological variables (Figure 12) or $CO_2$ (Figure 15). The $CO_2$ differences between the models and the observations may be caused by $CO_2$ flux or boundary condition errors, the two components impacting the modeled $CO_2$ mole fractions in addition to atmospheric transport. The cause of the total difference cannot be determined from the $CO_2$ data alone. The increased daily variance in $CO_2$ resulting from the ensemble calibration

process is shown in Figure 14. The 8-member ensemble often has the maximum $CO_2$ variance. Table 5 shows the spread (model-ensemble mean) and RMSE (model-data) ratio of the $CO_2$ mole fraction for the full and calibrated 10-member ensemble at each in-situ $CO_2$ observation tower. The ratio of the variances is an estimate of the contribution of the transport errors to the $CO_2$ model data mismatch for the summer of 2008. This table shows that the transport errors represent about 20% to 40% of the $CO_2$ model-data mismatch. We found that values after calibration show a slight increase compared to the full

ensemble.

## 4. Discussion

### 4.1 Impact of calibration on ensemble statistics

The calibration of the multi-physics/multi-analysis ensemble using SA and GA optimization techniques generated 10-, 8- and 5- member ensembles with a better representation of the error statistics of the transport model than the initial 45-member

ensemble. One of our goals was to find sub-ensembles that fulfil the criteria of Section 2.7, independent of the selection algorithm and for multiple meteorological variables. Wind speed and wind direction statistics only improve by a modest amount in the calibrated ensembles as compared to the 45-member ensemble, while PBLH statistics, namely the flatness of the rank histogram, shows a significant improvement in the calibrated ensembles. The variance in the calibrated ensembles increased relative to the 45-member ensemble but the potential for improvement was limited by the spread in the initial

ensemble. Stochastic perturbations (e.g. Berner et al., 2009) could increase the spread of the initial ensemble, which, combined with the suite of model configurations, could better represent the model errors. Here, we limited the 45-member ensemble to mass-conserved, continuous flow (i.e., unperturbed) members that can be used in a regional inversion. Future work should address the problem of using an under-dispersive ensemble before the calibration of the ensemble.

### 4.2 Single-variable and multiple-variable ensembles

We first attempted to calibrate the ensemble for each meteorological variable (i.e., wind speed, wind direction and PBLH). Table 3 shows that the different sub-ensembles were able to follow the criteria presented on Section 2.7, but the calibration of the single-variable ensembles did not allow us to find a unique sub-ensemble that can be used to represent the errors of the three variables. Therefore, the joint optimization of the three variables was required to identify an ensemble that best represents model errors across the three variables. By minimizing the sum of the squared rank-histogram scores of the three variables,

the selection algorithm found common solutions at the expense of less satisfactory rank histogram scores than were obtained for single-variable ensembles (cf. Table 4). We assumed that each variable was equally important to the problem, an



assumption that has not been rigorously evaluated. Future work on the relative importance of meteorological variables on $CO_2$ concentration errors would help weigh the scores in the selection algorithms.

### 4.3 Resolution and reliability

The calibrated ensembles show the rank histogram score closer to one (Table 4), that is, flatter rank histograms (Figure 9) compared to the 45-member ensemble (Table 2 and Figure 7). The sub-ensembles do have a greater variance than the large ensemble (i.e., improved reliability) (Figure 14). However, the spread-skill relationship (i.e., resolution) of the calibrated ensembles do not show any major improvement compared to the 45-member ensemble, implying that the spread of the ensemble does not represent the day-to-day transport errors well. The disagreement between the rank histogram and the spread-

skill relationship suggests that using the score of the rank histogram alone may not be sufficient to measure the reliability of the ensemble (Hamill, 2001). Down-selection of ensembles has been implemented in other studies (e.g., Garaud and Mallet, 2011; Lee et al., 2016) but resolution is usually excluded from the calibration process similar to our study (no skill score optimization). To represent daily model errors, additional metrics should be introduced and the initial ensemble should offer a sufficient spread, possibly with additional physic parameterizations, additional random perturbations, or modifications of the

error distribution of the ensemble (Roulston and Smith, 2003).

### 4.4 Error correlations

Rank histograms, as explained in Section 2.3.1, evaluate the ensemble by ranking individual observations in a relative sense. The ensembles calibrated using the rank histograms may be representing the variances over the region correctly but not the covariances (Hamill, 2001). In this study, the calibrated ensembles show an improvement in the meteorological variances and

an increase in the $CO_2$ variances relative to the uncalibrated ensemble, but spatial structures of the errors (i.e., correlations) may have limited statistical sampling of the model error structures due to the limited number of ensemble members. For example, ensemble model prediction systems use at least 50 members to avoid sampling noise. Figure 15 shows the spatial correlation of 300 m DDA $CO_2$ errors with respect to the Round Lake site on DOY 180. We suspect here that reduced-size ensembles are impacted by sampling noise which would require additional filtering. Previous studies have suggested objective

methods to filter the noise in small-size ensembles (i.e., Ménétrier et al., 2015) or modeling the error structures using the diffusion equation (e.g., Lauvaux et al., 2009). Future work should address the impact of the calibration on the error structures as this information is critical in the observation error covariance to assess the inverse fluxes. Concerning the magnitudes of the error correlation, the calibrated sub-ensembles exhibit a larger contrast in correlation values compared to the 45-member error correlations. Overall, the different ensembles show similar flow-dependent spatial patterns which demonstrates that the

calibration process, even if generating sampling noise, preserves the dominant spatial patterns in the error structures. Therefore, the calibrated ensemble is likely to provide a better representation of the variances and a similar spatial error structure for the construction of error covariance matrices in regional inversions.



## 5 Conclusion

We applied a calibration (or down-selection) process to a multi-physics/multi-analysis ensemble of 45 members. In this calibration process, two optimization techniques were used to extract a sub-set of members from the initial ensemble to improve the representation of transport model errors in $CO_2$ inversion modeling. We used purely meteorological criteria to calibrate the

ensemble and avoid contaminating the calibration with $CO_2$ flux errors. The calibrated ensembles were optimized using criteria based on the flatness of the rank histogram. We generated different calibrated ensembles for three meteorological variables; PBL wind speed, PBL wind direction and PBLH. With these techniques, we identified sub-ensembles by calibrating the three variables jointly. Both techniques show that calibrated small-size ensembles can reduce the score of the rank histogram flatness and therefore improve the representation of the model error variances with few members (between 5 and 10 members).

The calibration techniques improved the spread (flatness of the rank histogram) of the ensembles, and slightly improved the biases, which were already small in the larger ensemble, but the calibration did not improve daily atmospheric transport errors as shown by the spread-skill relationship. We assessed how the calibrated ensemble errors propagate into the $CO_2$ mole fractions simulated with identical $CO_2$ fluxes (i.e., independent of the atmospheric conditions). The spread from the calibrated ensembles represented from 20% to 40% (Table 5) of the model-data 300 m DDA $CO_2$ mismatches for summer 2008. These

results suggest that additional errors in $CO_2$ fluxes and/or large-scale boundary conditions represent a large fraction of the differences between modeled and observed $CO_2$. Error correlations of the calibrated ensembles were compared to the large ensemble to identify any impact of the calibration. Compared to the initial error structures, the calibrated ensembles are most likely affected by sampling noise across the region which suggest that additional filtering or modeling of the errors would be required in order to construct the error covariance matrix for regional $CO_2$ inversion.

*Code availability*. The code is accessible under request by contacting the corresponding author (lzd120@psu.edu).

*Data availability*. Meteorological data were obtained from the University of Wyoming's online data archive (http://weather.uwyo.edu/upperair/sounding.html) for the 14 rawinsonde stations. Tower Atmospheric CO2

Concentration data set is available on-line [http://daac.ornl.gov] from Oak Ridge National Laboratory Distributed Active Archive Center, Oak Ridge, Tennessee, USA. http://dx.doi.org/10.3334/ORNLDAAC/1202. The other two towers (Park Falls-WLEF and West Branch-WBI) are part of the Earth System Research Laboratory/Global Monitoring Division (ESRL/GMD) tall tower network (Andrews et al., 2014; https://www.esrl.noaa.gov/gmd/ccgg/insitu/). The WRF model results are accessible under request by contacting the corresponding author (lzd120@psu.edu).

*Author contribution*. L. Díaz Isaac performed the model simulations, calibration and the model-data analysis. The calibration technique was coded by M. Bocquet, L. Díaz-Isaac and T. Lauvaux based on the work of Garaud and Mallet (2011). T. Lauvaux, M. Bocquet and K. J. Davis provided guidance with the calibration and model-data analysis. All authors contributed to the design of the study and the preparation the paper.



## Acknowledgements

This research was supported by NASA's Terrestrial Ecosystem and Carbon Cycle Program, grant NNX14AJ17G, NASA's Earth System Science Pathfinder Program Office, Earth Venture Suborbital Program, grant NNX15AG76, NASA Carbon
5   Monitoring System, grant NNX13AP34G, and an Alfred P. Sloan Graduate Fellowship. We thank Dr. Natasha Miles, Dr. Chris E. Forest and Dr. Andrew Carleton for fruitful discussions. Meteorological data used in this work was provided by University of Wyoming's online data archive (http://weather.uwyo.edu/upperair/sounding.html). Observed atmospheric CO2 mole fraction was provided by NOAA Earth System Research Laboratory (cited in the text) and PSU in-situ measurement group are archive as:  Miles, N.L., S.J. Richardson, K.J. Davis, A.E. Andrews, T.J. Griffis, V. Bandaru, and K.P. Hosman.
10  2013. NACP MCI: Tower Atmospheric $CO_2$ Concentrations, Upper Midwest Region, USA, 2007-2009. Data set. Available on-line [http://daac.ornl.gov] from Oak Ridge National Laboratory Distributed Active Archive Center, Oak Ridge, Tennessee, USA. http://dx.doi.org/10.3334/ORNLDAAC/1202.

## Appendix A

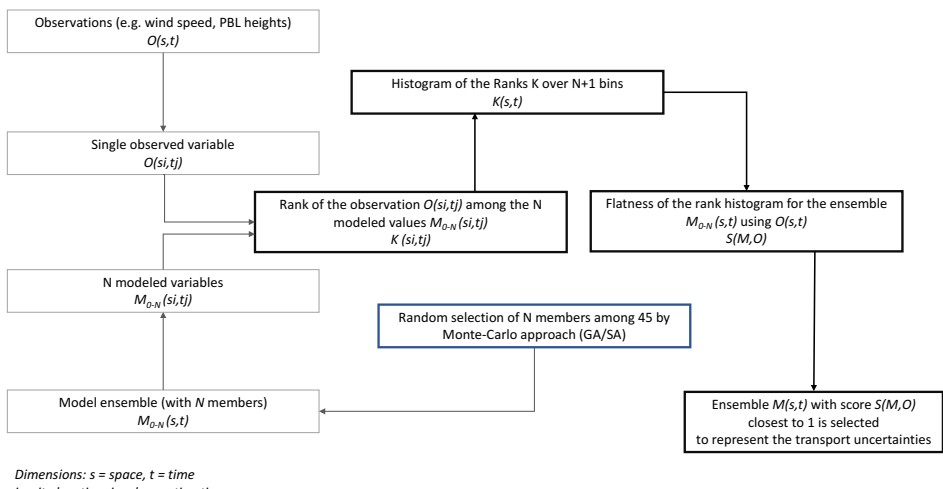

Figure A1. Diagram of the rank histogram process and selection of subensembles based on the rank histogram score.





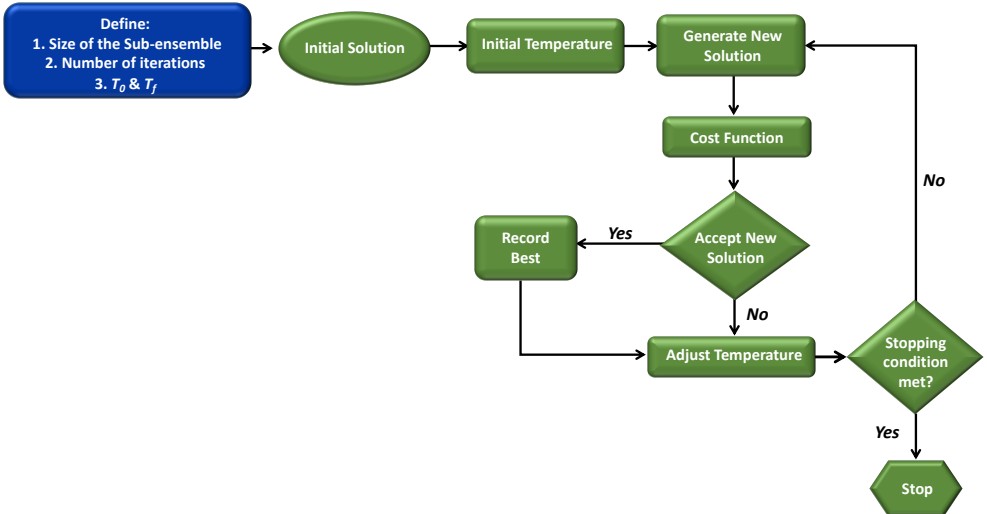

**Figure A2. Diagram of Simulated Annealing algorithm process.**

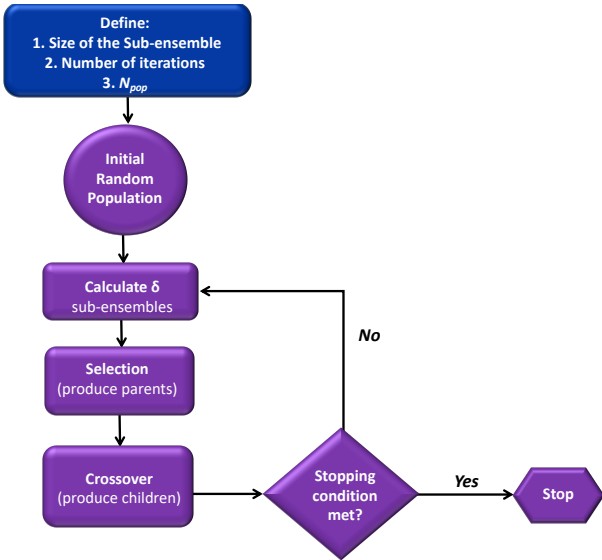

**Figure A3. Diagram of the Genetic Algorithm.**





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





**Table 1. Physics schemes used in WRF for the sensitivity analysis.**

| Parameter | Options |
|---|---|
| **Land Surface Model** | Noah LSM<br>Rapid Update Cycle (RUC) LSM<br>5-layer Thermal Diffusion |
| **Planetary Boundary Layer (PBL) scheme** | Yonsei University (YSU)<br>Mellor-Yamada-Janjic (MYJ)<br>Mellor-Yamada-Nakanishi-Niino Level 2.5 (MYNN2.5) |
| **Surface Layer** | MM5 similarity<br>Eta Similarity<br>MYNN surface layer |
| **Cumulus** | Kain-Fritsch (KF)<br>Grell-3Devenyi (G3D)<br>No cumulus parameterization |
| **Microphysics** | WSM 5-class<br>Thompson et al., (2004) |
| **Shortwave/Longwave radiation physics** | Dudhia/Rapid Radiative Transfer Model (RRTM) |
| **Initial & Boundary Conditions** | North America Regional Reanalysis (NARR)<br>Global Final Analysis (FNL) |

**Table 2. Rank histogram score ($\delta$), biases and standard deviation ($\sigma$) of the 45-member ensemble for wind speed, wind direction and PBLH computed across 14 rawindsonde sites using daily 0000 UTC observations for June 18 to July 21 of 2008 in the upper Midwest of the U.S.**

| Variables | $\delta$ | Bias | $\sigma$ |
|---|---|---|---|
| Wind Speed | 6.1 | 0.7 m/s | 3.5 m/s |
| Wind Direction | 6.2 | -0.6 degrees | 55.7 degrees |
| PBLH | 3.2 | 98.2 m | 787.5 m |

**Table 3. Calibrated ensembles generated by both SA and GA and their rank histograms scores and bias for each variable.**

| N | Variable | Sub-Ensemble | $\delta$ | Bias |
|---|---|---|---|---|
| **10** | **WSPD** | [5 13 14 16 17 29 33 35 39 45] | 3.8 | 0.4 m/s |
| | **WDIR** | [5 13 14 16 17 20 31 33 34 37] | 3.4 | -0.6 deg. |
| | **PBLH** | [2 11 14 23 27 31 35 37 43 44] | 0.4 | 58 m |
| **8** | **WSPD** | [11 14 16 31 35 37 39 45] | 3.7 | 0.5 m/s |
| | **WDIR** | [14 15 17 20 23 33 34 37] | 3.9 | -1 deg. |
| | **PBLH** | [12 13 14 23 26 28 37 44] | 0.8 | 75.5 m |
| **5** | **WSPD** | [5 14 29 36 39] | 3 | 0.4 m/s |
| | **WDIR** | [14 23 33 34 37] | 1.9 | 0.3 deg. |
| | **PBLH** | [2 5 13 31 44] | 0.1 | 69 m |





**Table 4. Ensemble members, rank histogram scores (δ), bias, and standard deviation (σ) for wind speed, wind direction and PBLH for the calibrated sub-ensembles generated with SA.**

| N | Sub-ensemble | Wind Speed | | | Wind Direction | | | PBLH | | |
|---|---|---|---|---|---|---|---|---|---|---|
| | | δ | Bias m/s | σ m/s | δ | Bias Deg. | σ Deg. | δ | Bias m | σ m |
| 10 | [14 17 23 26 28 33 34 35 37 45] | 5.5 | 0.6 | 3.6 | 4.6 | -0.6 | 58 | 1.5 | 79.7 | 817.4 |
| 8 | [5 6 14 17 26 33 34 37] | 5.6 | 0.6 | 3.6 | 3.4 | -0.7 | 58.5 | 1.6 | 71.8 | 823.4 |
| 5 | [16 17 23 33 35] | 5 | 0.5 | 3.6 | 3.4 | -0.7 | 59 | 0.6 | 76.2 | 810.7 |

**Table 5. Spread (model-ensemble mean), RMSE (model-data) and ratio (Spread$^2$/RMSE$^2$) at each of the in-situ CO$_2$ mixing ratio towers, for the 45-member ensemble and 10-member ensemble calibrated with SA and GA.**

| Sites | 45-Member Ensemble | | | SA 10-Member Ensemble | | | GA 10-Member Ensemble | | |
|---|---|---|---|---|---|---|---|---|---|
| | Spread (ppm) | RMSE (ppm) | Ratio (%) | Spread (ppm) | RMSE (ppm) | Ratio (%) | Spread (ppm) | RMSE (ppm) | Ratio (%) |
| Centerville | 4.3 | 9.3 | 19.1 | 4.7 | 9.6 | 22.7 | 4.4 | 9.4 | 20.4 |
| Galesville | 5.8 | 10.4 | 28 | 5.5 | 9.9 | 28.2 | 5.4 | 9.6 | 29.3 |
| Kewanee | 5.2 | 8.5 | 35.8 | 4.6 | 8 | 29.1 | 4.7 | 8.1 | 31.2 |
| Mead | 5.1 | 9.4 | 23.7 | 5 | 9.1 | 23.3 | 4.8 | 9 | 20.9 |
| Round Lake | 4.5 | 10.8 | 16 | 4.6 | 10.5 | 16.7 | 4.6 | 10.4 | 16.4 |
| WBI | 5.4 | 9.4 | 35.6 | 5.4 | 9.1 | 37.5 | 5.5 | 9.2 | 37.7 |
| LEF | 4.6 | 7.5 | 37.7 | 5.1 | 8.1 | 40 | 5.1 | 8.3 | 40.1 |


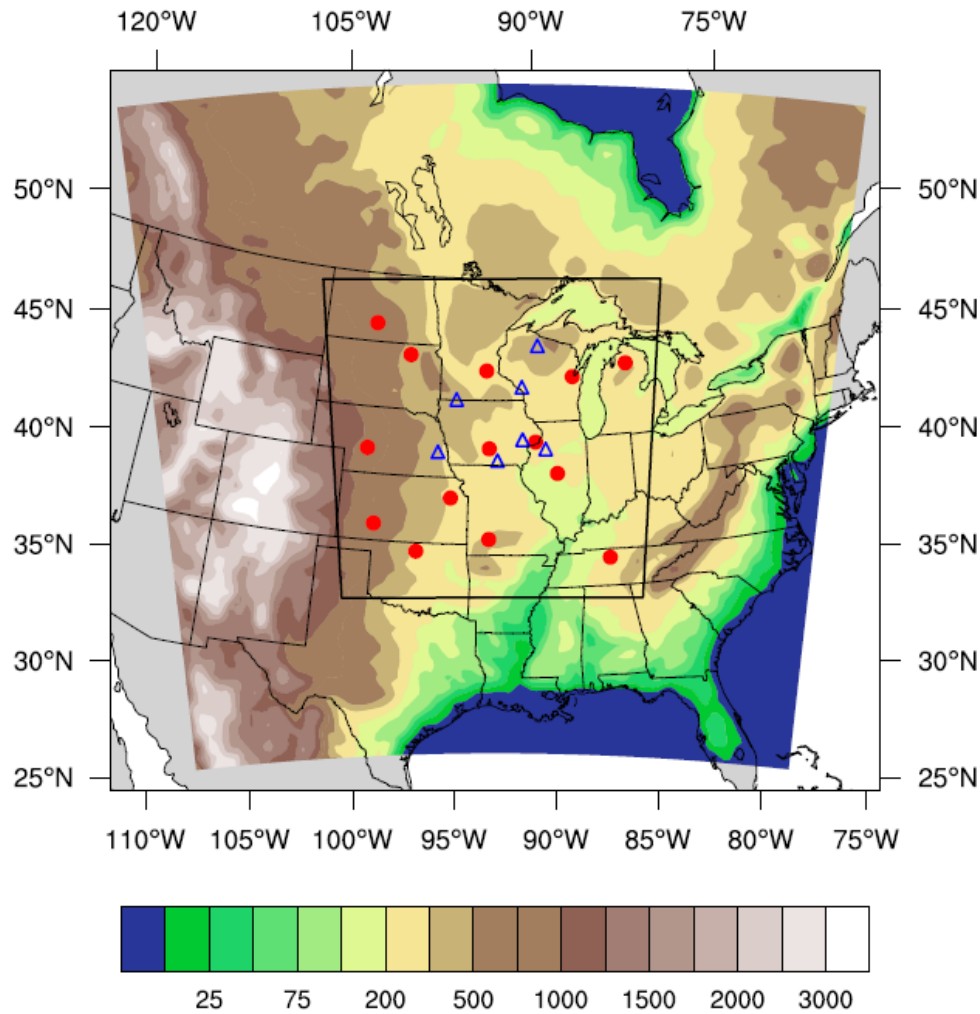

**Figure 1: Geographical domain used by WRF-ChemCO$_2$ physics ensemble. The parent domain (d01) has a 30-km resolution, the inner domain (d02) has a 10-km resolution. Contours represent terrain height in meters. The inner domain covers the study region and includes the rawinsonde sites (red circles) and the CO$_2$ towers (blue triangles) locations.**



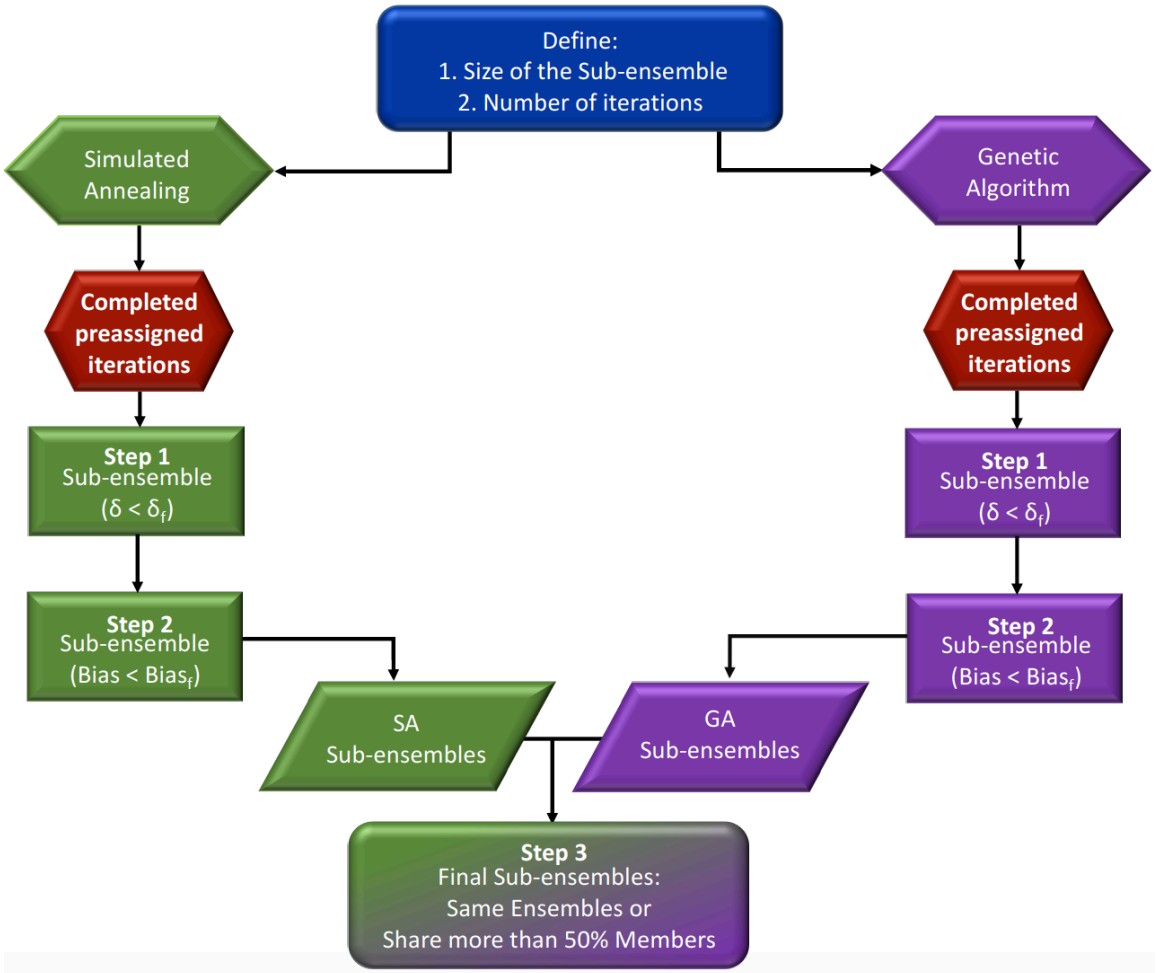

**Figure 2. Diagram of the process of selection of reduced-sized ensembles explained on section 2.7. In this diagram the sub-ensemble we show our two main thresholds after running each algorithm, sub-ensemble score has to be smaller than the full ensemble ($\delta < \delta_f$) and the sub-ensemble bias is smaller than the full-ensemble bias (Bias < Bias$_f$).**





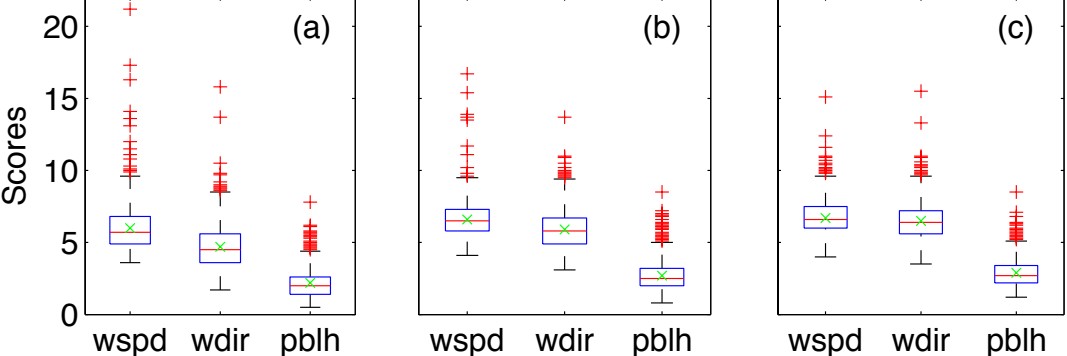

**Figure 3. Box plot of the rank histogram scores of the different sub-ensembles of 10 (a), 8(b), and 5 (c) members accepted by the SA. Each figure shows the rank histograms scores for the different variables PBL wind speed (wspd), PBL wind direction (wdir) and PBLH. The top of the box represents the 25$^{th}$ percentile, the bottom of the box is the 75$^{th}$ percentile, the red line in the middle is the median and the green 'x' the mean. Outliers beyond the threshold values are plotted using the '+' symbol.**

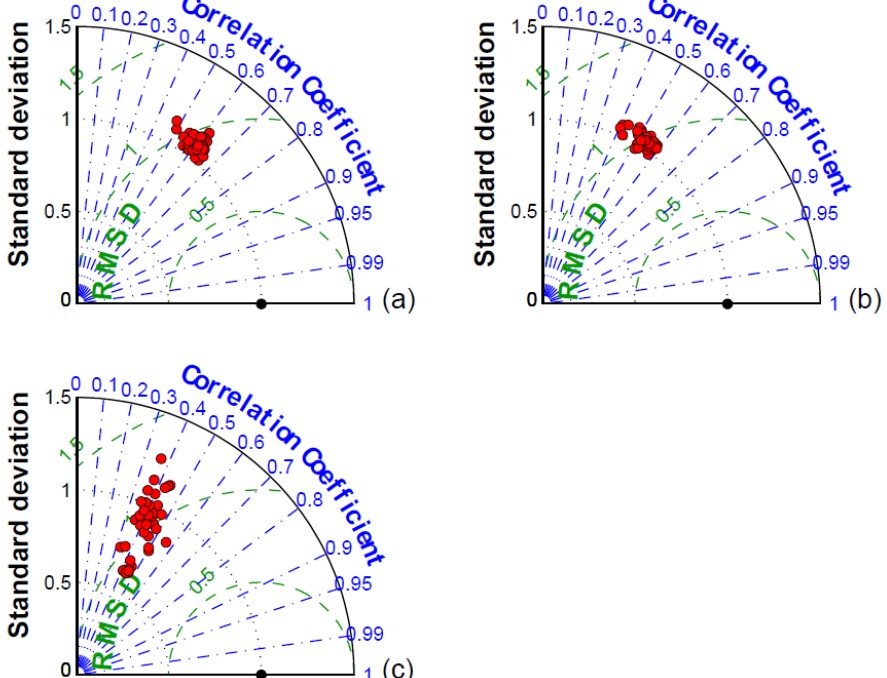

**Figure 4. Taylor diagram comparing the 0000 UTC rawinsonde observations (300 m wind speed (a), 300 m wind direction (b) and PBLH (c)) to the 45 model configurations (red circles).**




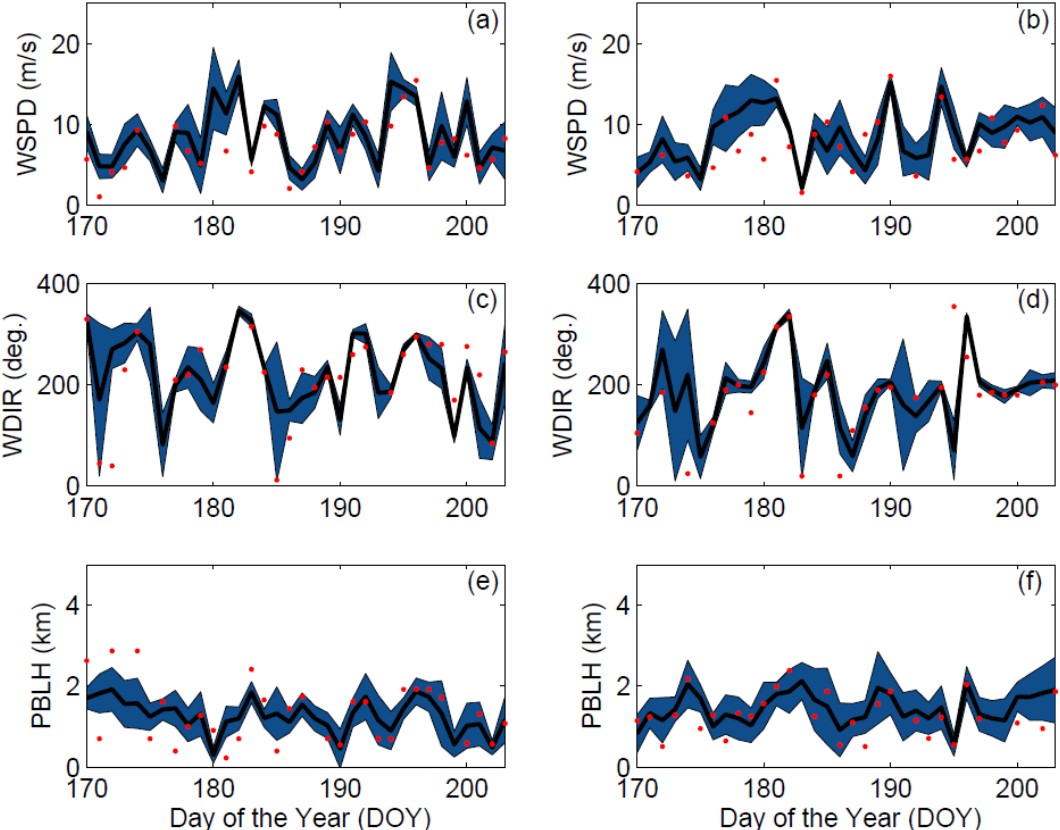

**Figure 5.** Time series of the simulated and observed for 300 m wind speed (a-b), 300 m wind direction (c-d) and PBLH (e-f) at GRB (a,c,e) and TOP (b,d,f) sites. The shaded blue area represents the spread (i.e. RMSD) of the full ensemble, the solid line the ensemble mean and the red dots the observations at 0000 UTC.





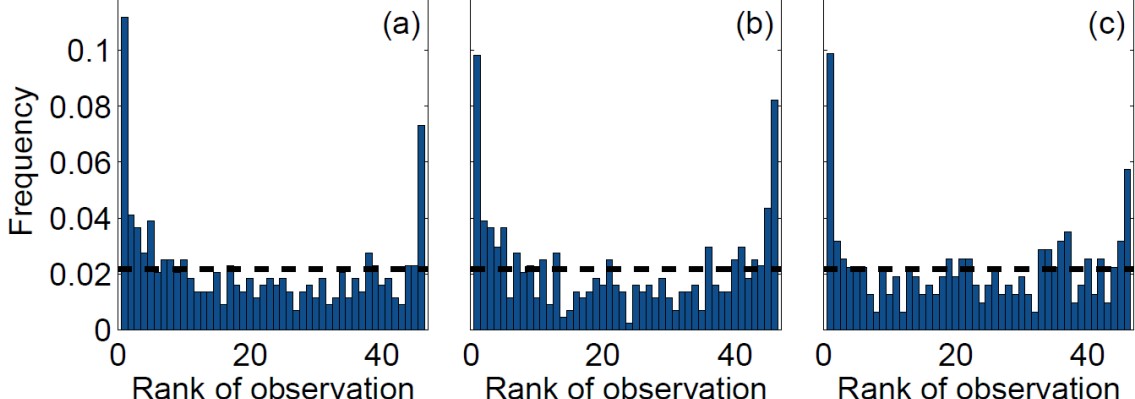

**Figure 6.** Rank histogram of the 45-member ensemble for wind speed (a), wind direction (b) and PBLH (c) using 14 rawinsonde sites available over the region. The horizontal dashed line ($\bar{r}$) corresponds to the ideal value for a flat rank histogram with respect to the number of members.





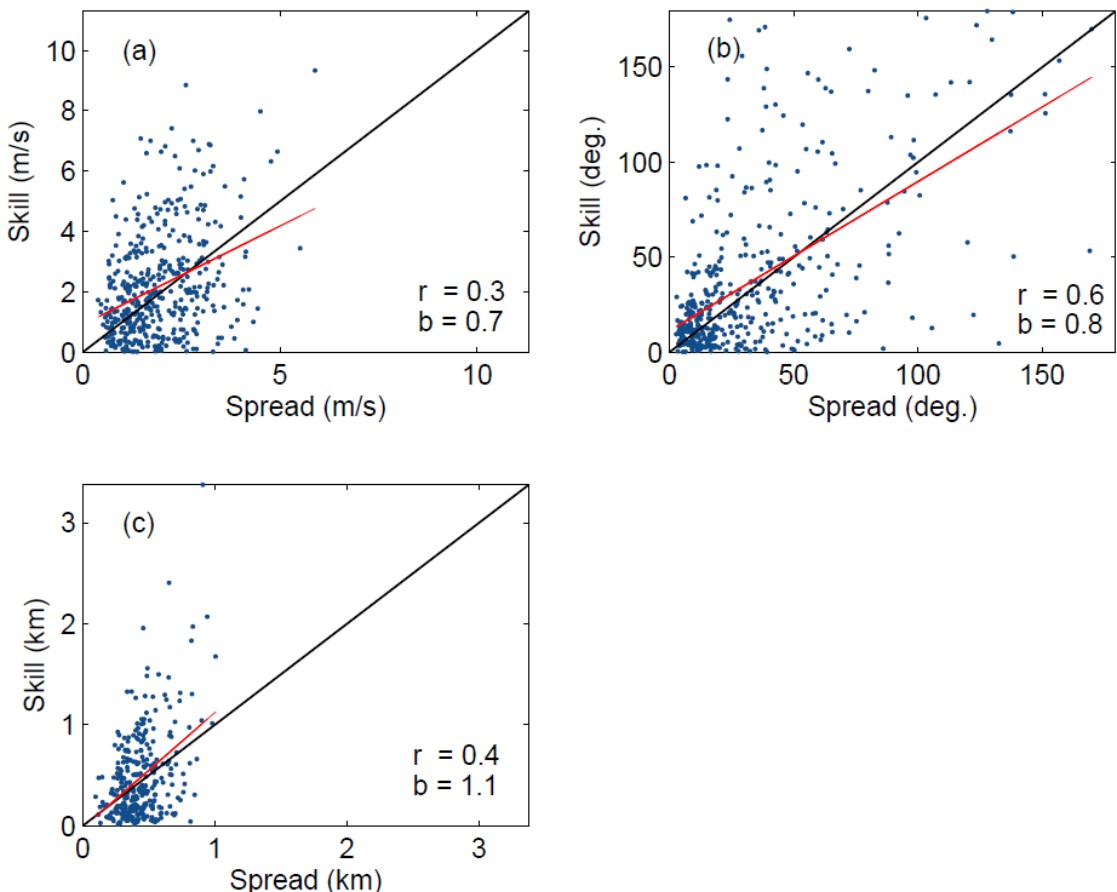

**Figure 7. Spread-skill for (a) wind speed, (b) wind direction and (c) PBLH using the 14 rawinsonde sites available over the region. Each point represents the model ensemble spread (standard deviation of the model-data difference) and skill (mean absolute error) for each observation. A one-to-one line is plotted in black and a line of best fit is plotted in red. Correlation (r) and slope (b) of the line of best fit of the spread-skill relationship are plotted as well.**



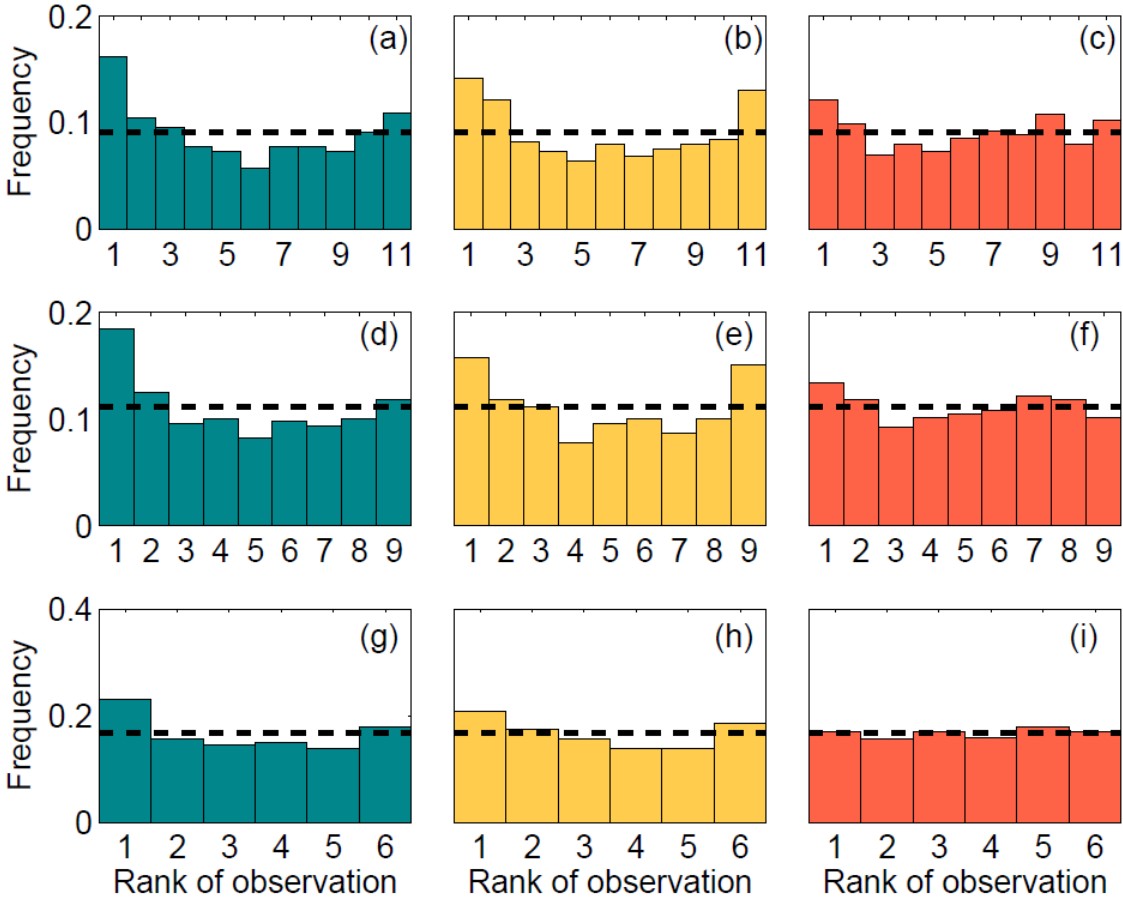

**Figure 8. Rank histograms of the calibrated ensembles found for wind speed (a, d, g), wind direction (b, e, h) and PBLH (c, f, i) for each of the ensemble size. The upper, middle and lower panels correspond to the ensemble with 10, 8, and 5 members, respectively. The horizontal dashed line ($\bar{r}$) corresponds to the ideal value for a flat rank histogram with respect to the number of members.**



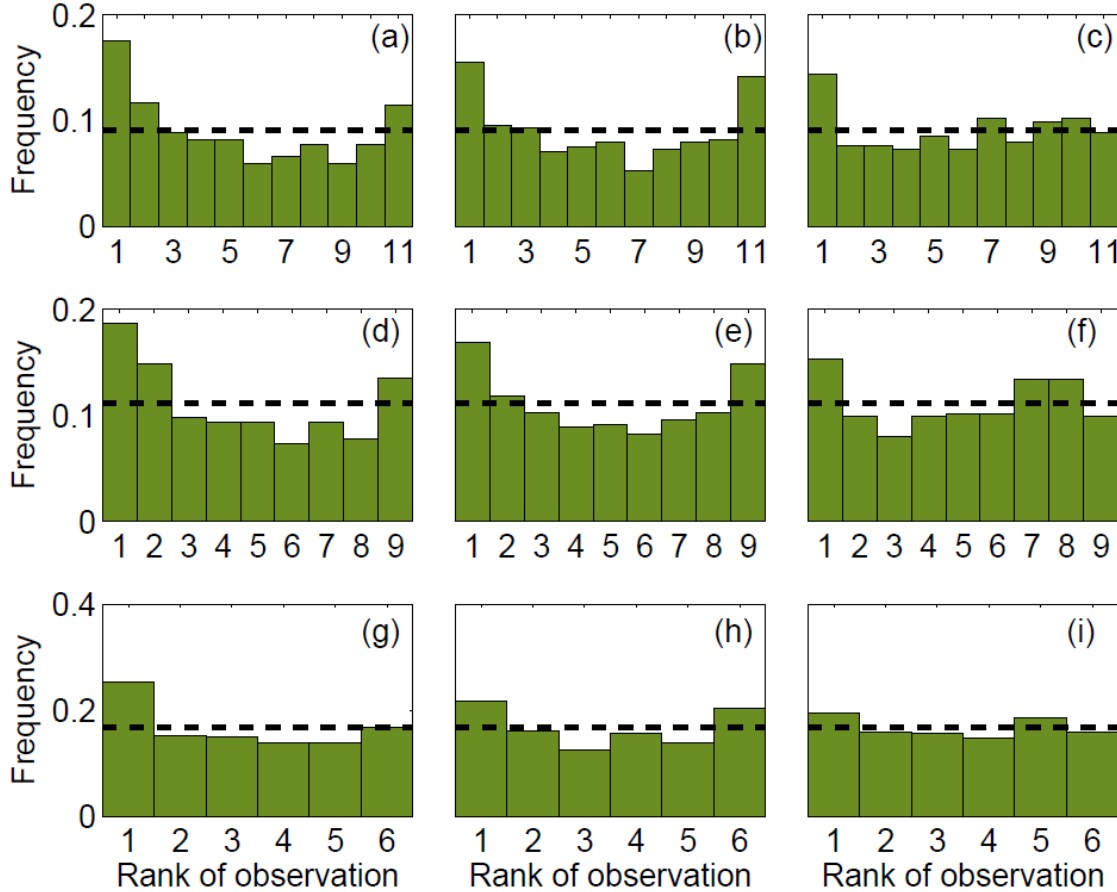

**Figure 9.** Rank histograms of wind speed a, d, g), wind direction (b, e, h) and PBLH (c, f, i) using the calibrated ensembles found with SA. The upper, middle and upper lower panels correspond to the ensemble with 10, 8 and 5 members, respectively. The horizontal dashed line ($\bar{r}$) corresponds to the ideal value for a flat rank histogram with respect to the number of members.




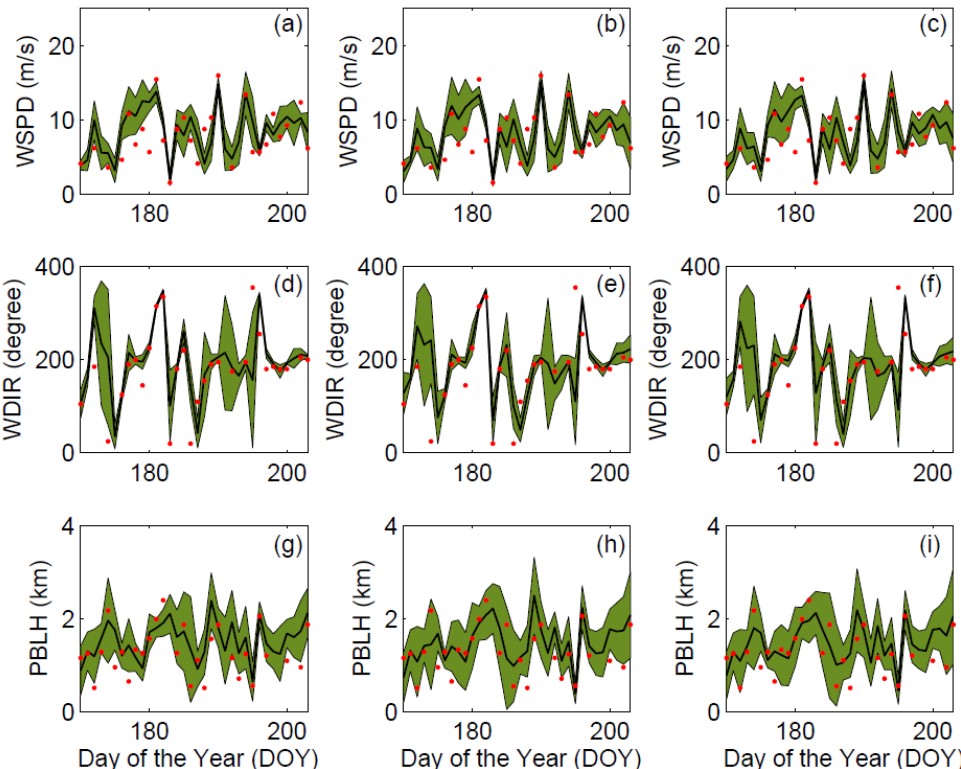

**Figure 10. Time series of simulated and observed 300 m wind speed (a-c), 300 m wind direction (d-f) and PBLH (g-i) using the 5 , 8- and 10-member calibrated ensembles (first, second and third column respectively) at the TOP rawinsonde site. The green shaded area represents the spread (i.e., Root Mean Square Deviation) of the ensemble, the black line is the mean of the ensemble and the red dots are the observations at 0000UTC.**





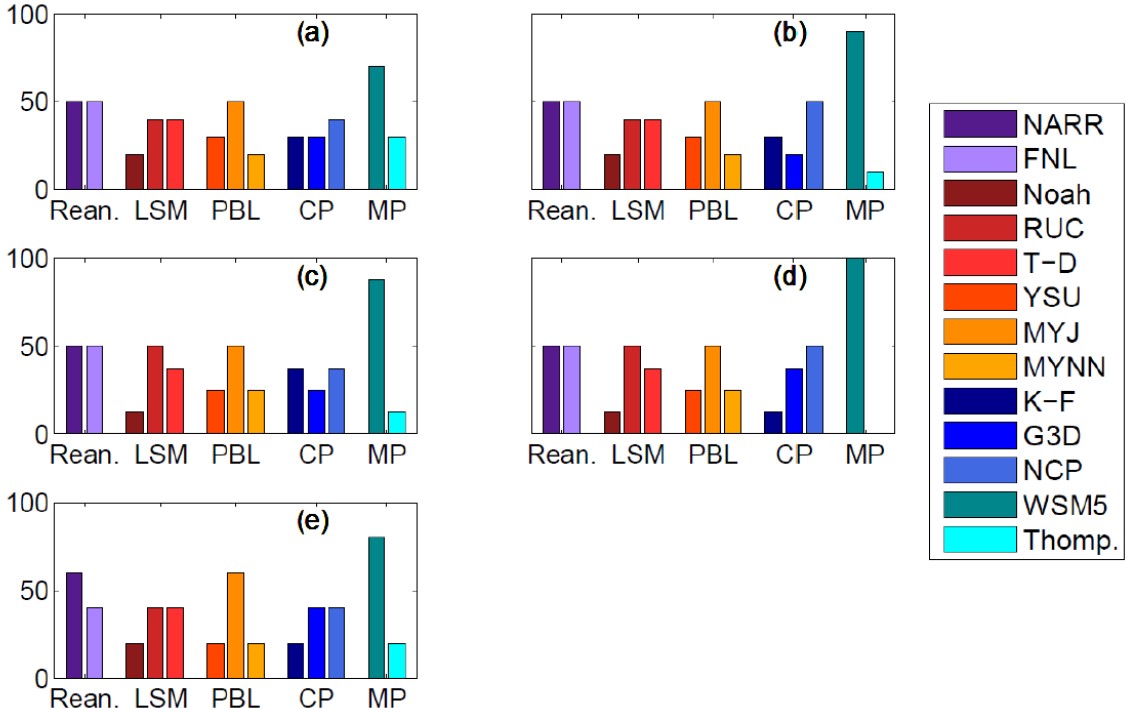

**Figure 11.** Frequency with which the physics schemes are used for the SA (a, c, e) and GA (b, d, e) calibrated ensembles of 10 members (a-b), 8-members (c-d) and 5-members (e).

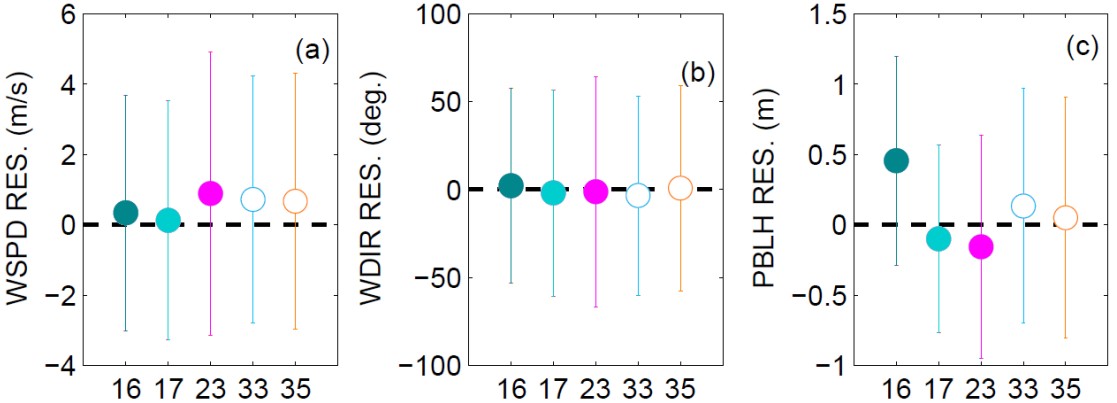

5  **Figure 12.** Residual (model-data mismatch) mean and standard deviation of individual members for wind speed (a), wind direction (b), PBLH (c) using the SA and GA calibrated sub-ensemble of five members.





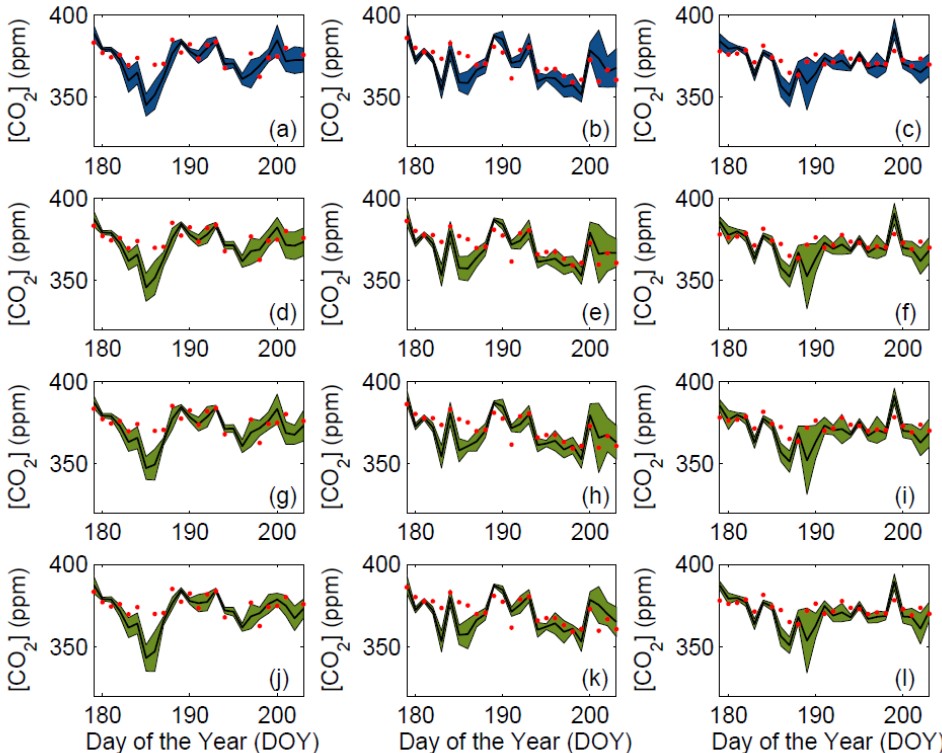

**Figure 13. Ensemble mean and spread (i.e., RMSD) of the daily daytime average (DDA) at approximately 100 m CO₂ concentrations at Mead (first column a,d,g,j), WBI (middle column b,e,h,k) and WLEF (last column c,f,i,l) towers using SA calibrated ensembles. Rows from top to bottom are 45, 10, 8 and 5 member ensembles. The blue area is the spread of the 45-member ensemble, the green area is the spread is the spread of the calibrated (10-, 8- and 5-member) ensemble, the black line is the mean of the ensemble and the red dots are the observations.**



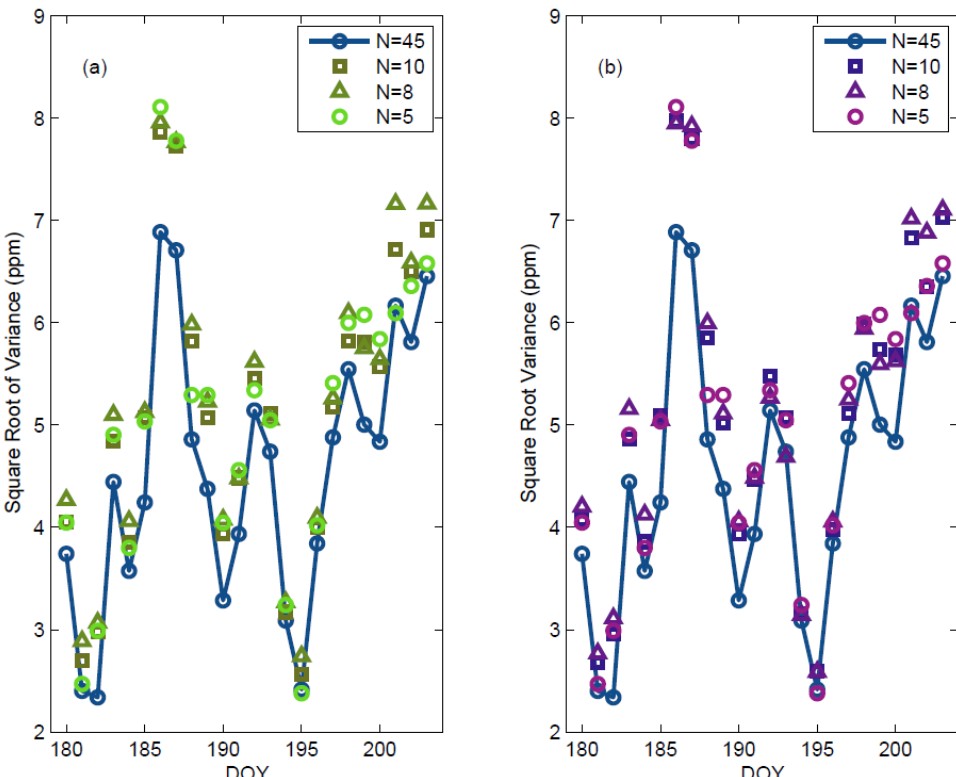

**Figure 14. Sum of the CO2 mixing ratio variance of the large ensemble (45-members) and the different sub-ensembles selected with the SA (a) and GA (b) down-selection techniques.**

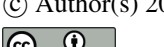


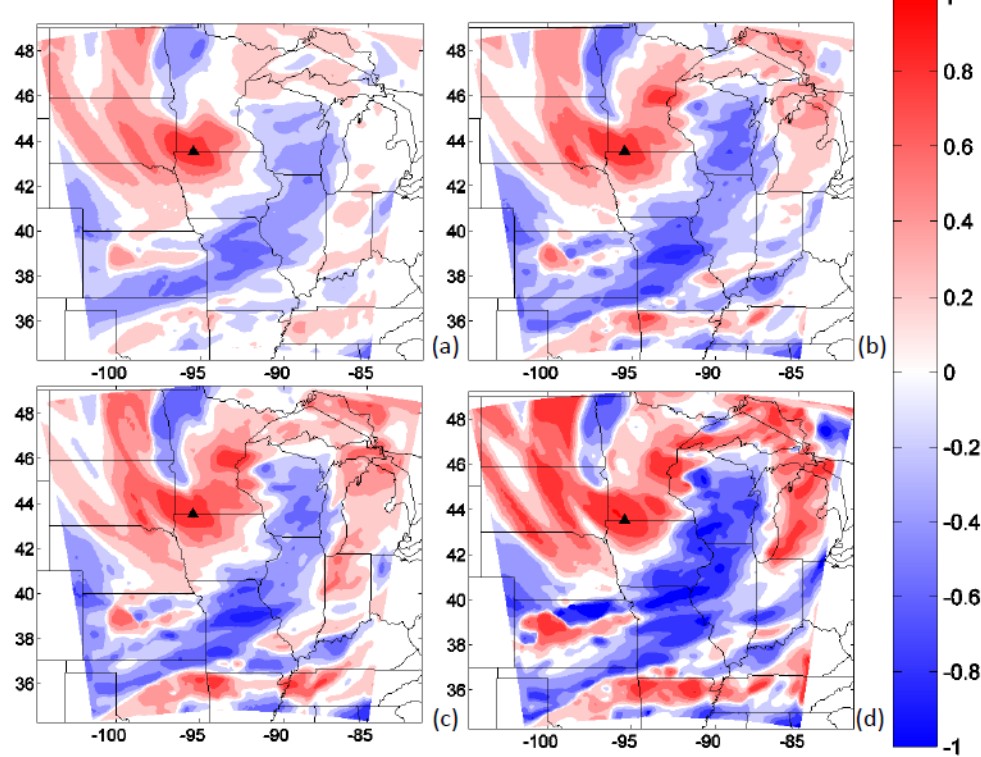

**Figure 15.** Spatial correlation of $CO_2$ for the 45- (a), 10-(b), 8-(c) and 5-member (d) ensembles with respect to the location of the Round Lake tower for DOY 180. This figure uses the calibrated ensembles of 10-, 8-, and 5-members found by the SA technique.