# Peer review of "Calibration of a multi-physics ensemble for estimating the uncertainty of a greenhouse gas atmospheric transport model"

_Atmospheric Chemistry and Physics, 2018_

## Referee Comment (RC1) · Enting (Referee) · 6 Jan 2019

Report on "Calibration of a multi-physics ensemble ....."

This is a significant study and is appropriate for publication in ACP. However there a few places where the terminology could be clarified.

Overall, i think the term "selection" or "down-selection" is preferable to "calibration " for the process that is being used.

Also, in many places, it would be better to replace "errors" with "uncertainty" (i.e. statistical characterisation of the unknown errors).

The flatness of the rank histogram is the primary criterion for selecting the ensembles. What doesn't seem to be discussed is the significance of various departures from flatness (as a function of the numbers of bins and the number of samples in the histogram). What fraction of the roughly 8 million possible 6 member ensembles (from the 45 cases) have essentially the same flatness. It is these questions that need to be clarified for understanding of whether the different results from SA vs GA are selecting from different populations of near optimal cases, or whether the differences are pretty much what you would expect from statistically based optimization on a population with a very flat minimum.

However, what I don't understand about the SA and GA searches is "why bother". Why not just look at all the cases explicitly (About 3 billion for the 10 member ensembles).

For M observations, all you need is a 45 by M table of the p_i ( i.e deviation between model and obs).

Then generate each sub ensemble in turn. For each case you scan the table and for each of the 45, you count the number where that model is part of your current sub ensemble AND p_i is less than zero. This number tells you which bin to increment. After dealing with all M observations, calculate delta and J As you work through all 3 billion possible sub ensembles, keep track of the best J (and note which ensemble) and any other statistics that you want .

This looks like it is well within the capabilities of modern computers.

For many purposes, there is no need to store stuff about all 3 billion ensembles, but if you wanted to, you could store all the J values in about the same amount of memory that I have on the sd card in my low-end smart phone.

As a minor point of notation, the ensemble, defined as a set, S, is indicated by upright font when it is introduced (p8, L27) but is shown as a slant font (as used for algebraic variables) as is done on the next line, and in eqn 3 and most later places. The usage

should be made consistent. Also, subscripts that are words or abbreviations of words, upright font should be used.

Ian Enting.
* * *

---

## Referee Comment (RC2) · Bruhwiler (Referee) · 11 Jan 2019

This is a very interesting study that seems to make some progress on an important issue for atmospheric inversions - how can we estimate atmospheric transport uncertainties? Most of us just use educated guesses, so it's really nice to be shown a potential way to do better even if it appears to be a lot of work and computational expense. I think the paper should be useful to the community of "flux inversers". One slightly disappointing thing is that $CO_2$ BC errors cannot be distinguished from transport errors making me look forward to trying this with a global model.

I have mainly minor comments, and there are a few things I didn't follow and would like

to better understand.

Abstract, L19 - I think "observations" should be added to the beginning of this sentence.

P2, L1- On what basis do these studies rule out spatial scale as a factor in inversion differences? Some of these studies use results from models with different spatial resolutions.

P2, L30 - The measurement people would object to the use of "ppmv" rather than "ppm" here because CO2 deviates from being an ideal gas. ppmv also appears elsewhere.

P5, L25 and throughout - The v in the for the virtual potential temperature gradient should be subscript to avoid confusion with a product.

P5, L26 - How robust is this definition for the PBLH? Is there a reference discussing this?

P6, L11- There's an extra "s" after rank.

P6, L20-25 - Would it be better to describe an under-dispersive ensemble as a distribution that is sharply peaked and shows less variability than observed? This would match up with the description of over-dispersive as having too much variability. Just a minor point though, I had tor read the sentence a couple of times, but then understood it.

P6, Eqn 1 - Is N the number of ensemble members and is this the same as "the number of models"? Also, it could be noted that the expectation is obs. evenly distributed over bins.

P7, L1 - Where does our statistical expectation of how well the ensemble matches the observed variability come from? Suppose that $r_j = \bar{r}$ in equation 1, then it seems that the model is getting the observed variability right, but what helps us to decide that this is overconfidence and not an extremely successful model?

P7, L4 - Does "samples" in this sentence refer to ensemble members or observations?

If covariances are underestimated, would this mean that there is nonindependent data and over-representation in a certain bin?

P7, L9- I think "mismatches" should not be plural here.

P7, L17-19 - Check the grammar here, "These" appears twice.

P8, L26 - The "flatness score" is the rank histogram score? Should stick with same terminology if possible.

P8, L27 - Is this N the same as the N that was used previously (e.g. the number of ensemble members)? I think this must be a different N that is something less than the previous one.

P8, L28 - It seems like a new symbol is being used for the rank histogram score here (it is delta in eqn 1). Is this because it's going to be optimized by the SA/GA procedures and so a cost function will be defined?

P9, L9-21 - I have a few questions about this description. First, isn't the deviation of delta from 1 what is being optimized here? I don't see how this is explicit in the notation. The other question I have is about the size of the sub-ensemble. Can the procedure test sub-ensemble sizes all of the way to N-1 and all of the way down to some minimum number, maybe 2?

P9, L30-31 - Is mutation a separate step here? Or is is considered part of "crossover"?

P10, L103 - I have the same question that I had for the SA, are the sizes of sub-ensembles allowed to vary?

P12, Section 3.2 - Does this answer my question about exploring the sizes of the sub-ensembles? One uses the largest frequency from the rank histogram and since this happens to be the first box, then than one gets used? Why are 5-member ensembles used?

P16, L7-9 - I'm struggling with the implication of this statement. It means that even

though the sub-ensemble has the right spread it doesn't mean the simulation will encompass the true values? What about bias? If the model is biased one could get this situation, right?

P16, Section 4.4 - I'm not sure I follow this argument. I see from Fig 15 that the spatial correlations of $CO_2$ get closer to 1 or -1, but I'm not sure why this happens with fewer ensemble members. It's stated that this is because of sample size (i.e. number of realizations?) but why should this result in a more intense correlation pattern? I would like to understand this.

---

## Author Comment (AC1) · 26 Mar 2019

**Answers to Referee Ian Enting comments:** *Review of Calibration of a multi-physics ensemble for greenhouse gas atmospheric transport model uncertainty estimations*

We thank the referee for the helpful comments that will improve the manuscript. In the text below, we have tried our best to respond to all the general and specific comments provided by the reviewer.

**Comments to Author:**
This is a significant study and is appropriate for publication in ACP. However there a few places where the terminology could be clarified.

**REF-C1:** Overall, I think the term "selection" or "down-selection" is preferable to "calibration " for the process that is being used.
**Author-C1:** We agree with the reviewer that our method is basically a selection of ensemble members to create an optimal ensemble. But beyond the simple selection of members, it also improves the representation of errors by calibrating the ensemble against actual meteorological data. The terminology is commonly used in the weather forecasting community, from which our technique was first applied in the early 90's. Considering the history of the terminology and the better representation of ensemble statistics, we decided to clarify in the abstract and the introduction for the broader audience but we kept the term "calibration" to preserve the idea of the regularization of our statistics in the later sections.

Abstract, P1, L20: *"Two **optimization techniques (i.e., simulated annealing and a genetic algorithm)** are used for the selection of the optimal ensemble using the flatness of the rank histograms as the main criterion."*

P4, L7-18: *"In this study, we start with a large multi-physics/multi-analysis ensemble of 45-members presented in Díaz-Isaac et al. (2018) and apply a down-selection or calibration process similar to the one explained in Garaud and Mallet (2011). Two principal features characterize an ensemble: reliability and resolution. The reliability is the probability that a simulation has of matching the frequency of an observed event. The resolution is the ability of the system to predict a specific event. Both features are needed in order to represent model errors accurately. **Our main goal is to down-select the large ensemble to generate a calibrated ensemble that will represent the uncertainty of the transport model with respect to meteorological variables of most importance in simulating atmospheric $CO_2$.** These variables are the horizontal mean PBL wind speed and wind direction, and the vertical mixing of surface fluxes, i.e. PBLH. We focus on the criterion that will measure the reliability of the ensemble, i.e. the probability of the ensemble in representing the frequency of events (i.e. the spatio-temporal variability of the atmospheric state). **For the down-selection of the ensemble, we will use two different techniques, simulated annealing and a genetic algorithm from now on refer as calibration techniques/process**. In a final step, the ensemble with the optimal reliability will be selected by minimizing the biases in the ensemble mean. We will evaluate which physical parameterizations play important roles in balancing the ensembles and evaluate how well a pure physics ensemble can represent transport uncertainty."*

**REF-C2:** Also, in many places, it would be better to replace "errors" with "uncertainty" (i.e. statistical characterization of the unknown errors).
**Author-C2:** Errors was replaced with uncertainties in some places or the manuscript.

**REF-C3:** The flatness of the rank histogram is the primary criterion for selecting the ensembles. What doesn't seem to be discussed is the significance of various departures from flatness (as a function of the numbers of bins and the number of samples in the histogram). What fraction of the roughly 8 million possible 6 member ensembles (from the 45 cases) have essentially the same flatness. It is these questions that need to be clarified for understanding of whether the different results from SA vs GA are selecting from different populations of near optimal cases, or whether the differences are pretty much what you would expect from statistically based optimization on a population with a very flat minimum.

**Author-C3:** The figure below (Figure A4) shows the frequency of the rank histogram scores for each calibration technique, sub-ensembles size and variable (wind speed, wind direction and PBLH). This is based on the sub-ensemble collected at the end of the process, where the rank histogram score and bias are smaller than the original 45-member ensemble. We found more cases in the lower scores for PBLH and in the higher scores for wind speed and wind direction. The figure shows that overall, wind speed controls a significant amount of the optimization because of the high frequency for large scores. However, wind speed doesn't impede the selection of ensembles with a small score for the other variables (PBLH and wind direction). We added this figure to the appendix and make some reference to this point in section 3.2.2:

Section 3.2.2, P13: "The rank histogram scores for all variables are greater than those for one-variable optimization (see Table 4). ***The high-rank histogram scores are associated with the equal weight gave to the three variables for this simultaneous calibration, where wind speed controlled the calibration process. For the calibration of the three variables together, we were not able to produce an ensemble for wind speed with a score smaller than four, this ends up limiting the selection of the calibrated ensemble for the rest of the variables (see Figure A4 in Appendix 1).*** In addition, all these calibrated sub-ensembles have biases …"

[Figure]

**Figure A4. Rank histogram score of calibrated sub-ensembles of different size generated by Simulated Annealing (a-c) and Genetic Algorithm (d-f). Each color bar represents the frequency of that scores for the three different variables wind speed (WSPD), wind direction (WDIR) and PBL height (PBLH).**

**REF-C4:** However, what I don't understand about the SA and GA searches is "why bother". Why not just look at all the cases explicitly (About 3 billion for the 10 member ensembles). For M observations, all you need is a 45 by M table of the p_i ( i.e deviation between model and obs). Then generate each sub ensemble in turn. For each case you scan the table and for each of the 45, you count the number where that model is part of your current sub ensemble AND p_i is less than zero. This number tells you which bin to increment. After dealing with all M observations, calculate delta and J As you work through all 3 billion possible sub ensembles, keep track of the best J (and note which ensemble) and any other statistics that you want. This looks like it is well within the capabilities of modern computers. For many purposes, there is no need to store stuff about all 3 billion ensembles, but if you wanted to, you could store all the J values in about the same amount of memory that I have on the sd card in my low-end smart phone.

**Author-C4:** We tested the brute-force solution at an early stage of the paper and concluded that the size of the sub-ensemble would become very rapidly a limitation. Beyond 10 members, the number of solutions increases very rapidly and requires hours if not days to compute. It is nearly impossible for 20 members or more. We note here that our objective was to use an objective

methodology applicable to any ensemble sizes. We have submitted a second study with a 25-member ensemble which, in this case, means 3,000 billion combinations, hence requiring our Monte Carlo approach.

**REF-C5:** As a minor point of notation, the ensemble, defined as a set, S, is indicated by upright font when it is introduced (p8, L27) but is shown as a slant font (as used for algebraic variables) as is done on the next line, and in eqn 3 and most later places. The usage should be made consistent. Also, subscripts that are words or abbreviations of words, upright font should be used.

**Author-C5:** We corrected the S and the subscripts as suggested. Also, we changed $J$ for $\delta$ to keep everything consistent throughout the article. Please see the next edited part:

"Both techniques generate a sub-ensemble ($S$) of size N. For the first test, we will use these algorithms to choose the combination of members that optimize the score of the reduced ensemble $\delta$ ($S$) (i.e., rank histogram score) for each variable. With this evaluation, we determine if each optimization technique yields similar calibrated ensembles, and if the calibrated ensembles are similar among the different meteorological variables. In the second test, we calibrate the ensemble for all three variables simultaneously, where we use the sum of the score squared: $[\delta\,(S)]^2$ :

$$[\delta(S)]^2 = [\delta_{\mathrm{wspd}}(S)]^2 + [\delta_{\mathrm{wdir}}(S)]^2 + [\delta_{\mathrm{pblh}}(S)]^2, \tag{3}$$

to control acceptance of the sub-ensembles. In Eq. (3), $\delta_{\mathrm{wspd}}(S)$, $\delta_{\mathrm{wdir}}(S)$ and $\delta_{\mathrm{pblh}}(S)$ are the scores of the sub-ensemble for PBL wind speed, PBL wind direction and PBLH respectively."

---

## Author Comment (AC2) · 26 Mar 2019

**Answers to Referee Lori Bruhwiler comments:** *Review of Calibration of a multi-physics ensemble for greenhouse gas atmospheric transport model uncertainty estimations*

We thank the referee for the helpful comments that will improve the manuscript. In the text below, we have tried our best to respond to all the general and specific comments provided by the reviewer.

**Comments to Author:**
This is a very interesting study that seems to make some progress on an important issue for atmospheric inversions - how can we estimate atmospheric transport uncertainties? Most of us just use educated guesses, so it's really nice to be shown a potential way to do better even if it appears to be a lot of work and computational expense. I think the paper should be useful to the community of "flux inversers". One slightly disappointing thing is that CO2 BC errors cannot be distinguished from transport errors making me look forward to trying this with a global model.
I have mainly minor comments, and there are a few things I didn't follow and would like to better understand.

**REF-C1:** Abstract, L19 - I think "observations" should be added to the beginning of this sentence.
**Author-C1**: Done.
*P1, L19: "Observed meteorological variables critical to inverse flux estimates, PBL wind speed, PBL wind direction and PBL height, are used to calibrate our ensemble over the region."*

**REF-C2:** P2, L1- On what basis do these studies rule out spatial scale as a factor in inversion differences? Some of these studies use results from models with different spatial resolutions.
**Author-C2:** The spatial scale is indeed an important factor to be considered for discrepancies among inversions. The text was modified as follows:
*P2, L1: "Large uncertainty and variability often exist among inverse flux estimates (e.g., Gurney et al., 2002; Sarmiento et al., 2010; Peylin et al., 2013; Schuh et al., 2013). These posterior flux uncertainties arise from varying spatial resolution, limited atmospheric data density ..."*

**REF-C3:** P2, L30 - The measurement people would object to the use of "ppmv" rather than "ppm" here because $CO_2$ deviates from being an ideal gas. ppmv also appears elsewhere.
**Author-C3**: We corrected the unit:
*P2, L30: "Approximately 3 ppm uncertainty in $CO_2$ mole fractions have been attributed to PBLH errors over Europe during the summer time (Gerbig et al., 2008; Kretschmer et al., 2012)."*
*P8, L8: "Transport model errors in atmospheric inversions are described in the observation error covariance matrix, hence in $CO_2$ mole fractions ($ppm^2$)."*

**REF-C4:** P5, L25 and throughout - The v in the for the virtual potential temperature gradient should be subscript to avoid confusion with a product.
**Author-C4:** We corrected the v of virtual potential temperature:
*P5, L25: "The PBLH was estimated using the virtual potential temperature gradient ($\nabla\theta_v$). The method identifies the PBLH as the first point above the atmospheric surface layer where (1) $\nabla\theta_v$ is greater than or equal to 0.2 K/km, and (2) the difference between the surface and the threshold level virtual potential temperature is greater than or equal to 3 K ($\theta_{vs} - \theta_v \geq 3K$)."*

**REF-C5:** P5, L26 - How robust is this definition for the PBLH? Is there a reference discussing this?
**Author-C5:** We use Seibert et al. (1999) and Seidel et al. (2010) to define the PBLH used in this study. However, evaluation of multiple vertical profiles from both simulations and radiosonde were used to explore the best technique and definition to define the PBLH height. The two definitions that we explored the most was the Bulk Richardson Number and the virtual potential temperature gradient. The Richardson number was showing a consistent underestimation of the PBLH. We identified the virtual potential temperature gradient as the most reliable algorithm to estimate PBLH. This evaluation relies on visual inspection of vertical potential temperature profiles, which may vary depending on expert judgement. However, for a lack of a better definition, we decided to use the virtual potential temperature gradient as the main definition of PBLH for both the model and the observation.

**Reference:**

Seibert, P., Beyrich, F., Gryning, S.-E., Joffre, S., Rasmussen, A., and Tercier, P.: Review and intercomparison of operational methods for the determination of the mixing height, Atmos. Environ., 34, 1001–1027, 1999.

Seidel D., A, C. O., and Li, K.: Estimating climatological planetary boundary layer heights from radiosonde observations: Comparison of methods and uncertainty analysis, J. Geophys. Res., 115, D16113, doi:10.1029/2009JD013680, 2010.

**REF-C6:** P6, L11- There's an extra "s" after rank.
**Author-C6:** Done
P6, L11: *"The criteria used for our down-selection process include rank histograms, rank histogram scores and ensemble bias."*

**REF-C7:** P6, L20-25 - Would it be better to describe an under-dispersive ensemble as a distribution that is sharply peaked and shows less variability than observed? This would match up with the description of over-dispersive as having too much variability. Just a minor point though, I had to read the sentence a couple of times, but then understood it.
**Author-C7:** We agree with the reviewer that a sharply peaked distribution may explain the lack of variability in the ensemble. However, an ensemble that is underdispersive may not only be affected by the lack of variability, but can also be affected by biases. We refer to Hamill (2001) who carefully explored the meaning of underdispersive ensembles. Rank histograms correspond to the evaluation of the ensemble for each observation, hence impacted by the spread but also the skill of the ensemble. We edited this part of the manuscript to avoid confusion:

P6, L20-25: *A rank histogram that deviates from the flat shape implies a biased, overdispersive or underdispersive ensemble.* ***A "U-shaped" rank-histogram indicates that the ensemble is underdispersive, normally in this type of ensembles the observations tend to fall outside of the envelope of the ensemble, this kind of histogram are associated with a lack of variability or an ensemble affected by biases (Hamill, 2001).*** *A "central-dome" (or "A-shaped") histogram indicates that the ensemble is overdispersive ..."*

**REF-C8:** P6, Eqn 1 - Is N the number of ensemble members and is this the same as "the number of models"? Also, it could be noted that the expectation is obs. evenly distributed over bins.

**Author-C8: :** Yes, "the number of models" is the number of ensemble members. We changed this to the number of members, in case this may cause confusion. See part of the edit to the sentence below.

P6, L29-30: *"and should ideally be close to 1 (Talagrand et al., 1999; Candille and Talagrand, 2005). In Eq.(1), N is the number of members (i.e., models)…"*

**REF-C9:** P7, L1 - Where does our statistical expectation of how well the ensemble matches the observed variability come from? Suppose that rj = r(bar) in equation 1, then it seems that the model is getting the observed variability right, but what helps us to decide that this is overconfidence and not an extremely successful model?

**Author-C9:** We agree with the reviewer that rank histograms alone cannot solve that problem. Our expectation is based on rj =r(bar), following equation 1. One way to evaluate if the ensemble is overconfident or extremely good is by combining the rank histograms to other statistical analyses. An overconfident ensemble shows an uncertainty (model-data mismatch) larger than the spread. Therefore, a statistical analysis that will give us more information about the spread and the uncertainty is the spread-skill relationship plot (see Figure 7). Figure 7 from the paper shows the spread-skill relationship of the three variables (i.e., wind speed, wind direction and PBLH) for the large ensemble. If we look at the PBLH spread-skill relationship (Figure 7c), the spread of the ensemble is smaller than the skill (uncertainty), this behavior also shows up when we calibrate our ensemble to a rank histogram that is 1 or below 1. Therefore, we were able to improve the rank histogram score of all the variables, especially PBLH getting close to 1 or lower, but the spread-skill relationship indicates that the spread is comparable to the skill of our ensemble. We note here that on a daily basis, the two quantities do not correlate which indicates a lack of resolution at fine time scales.

**REF-C10:** P7, L4 - Does "samples" in this sentence refer to ensemble members or observations? If covariances are underestimated, would this mean that there is nonindependent data and over-representation in a certain bin?

**Author-C10:** In this case the sample is the simulated variable at the different stations or grid points. Error correlations could lead to an over-representation of certain bins. Based on a more recent study currently under review in ACPD (https://www.atmos-chem-phys-discuss.net/acp-2018-1113/) and Figure 15 in the discussion section, our tower observations should remain independent thanks to the long distances between tower locations (>150km). But we fully agree that correlated observations would bias the histograms if the distance between the observations locations were smaller.

**REF-C11:** P7, L9- I think "mismatches" should not be plural here.

**Author-C11:** We change it to mismatch:

P7, L9: *"The bias, or the mean of the model-data mismatch, was used to assist the selection of the calibrated sub-ensemble."*

**REF-C12:** P7, L17-19 - Check the grammar here, "These" appears twice.

**Author-C12:** Done
P7, L17-19: "*These statistical analyses will be used to describe the performance of each member (standard deviations and correlations), ensemble spread (root mean square deviation) and error structures in space (error covariance), which will allow us to evaluate all the important aspects of an ensemble.*"

**REF-C13:**P8, L26 - The "flatness score" is the rank histogram score? Should stick with same terminology if possible.
**Author-C13:** We fix the terminology, to keep everything consistent.
P8, L26: "*In this study, SA and GA techniques will randomly search for the different combinations of members and compute the rank histogram score.*"

**REF-C14:** P8, L27 - Is this N the same as the N that was used previously (e.g. the number of ensemble members)? I think this must be a different N that is something less than the previous one.
**Author-C14:** Yes, this N is the same that was used previously, which represents the number of ensemble members. Throughout the paper N always represents the number of members or models used in each ensemble, regardless of the size.

**REF-C15:** P8, L28 - It seems like a new symbol is being used for the rank histogram score here (it is delta in eqn 1). Is this because it's going to be optimized by the SA/GA procedures and so a cost function will be defined?
**Author-C15:** We changed *J* to delta to keep everything consistent.

**Section 2.5**
For the first test, we will use these algorithms to choose the combination of members that optimize the score of the reduced ensemble $\delta(S)$ (i.e., rank histogram score) for each variable. With this evaluation, we determine if each optimization technique yields similar calibrated ensembles, and if the calibrated ensembles are similar among the different meteorological variables. In the second test, we calibrate the ensemble for all three variables simultaneously, where we use the sum of the score squared: $[\delta(S)]^2$:

$$[\delta(S)]^2 = [\delta_{\mathrm{wspd}}(S)]^2 + [\delta_{\mathrm{wdir}}(S)]^2 + [\delta_{\mathrm{pblh}}(S)]^2, \tag{3}$$

to control acceptance of the sub-ensembles. In Eq. (3), $\delta_{\mathrm{wspd}}(S)$, $\delta_{\mathrm{wdir}}(S)$ and $\delta_{\mathrm{pblh}}(S)$ are the scores of the sub-ensemble for PBL wind speed, PBL wind direction and PBLH respectively.

**Section 2.5.1**
To minimize the score $\delta$, only two transitions to the neighbours are possible. First transition, if the score of the neighbour sub-ensembles $\delta(S')$ is lower than the current sub-ensemble $\delta(S)$, then *S'* becomes the current sub-ensemble and a new neighbour sub-ensemble is generated. Second transition, if the score of the neighbour sub-ensemble $\delta(S')$ is greater than the current sub-ensemble $\delta(S)$, moving to the neighbour S' only occurs through an acceptance probability. This acceptance probability is equal to $exp(-\frac{\delta(S')-\delta(S)}{T})$ and it only allows the movement to the neighbor S' if $u < exp(-\frac{\delta(S')-\delta(S)}{T})$. For the acceptance probability, *u* is a random number uniformly drawn from [0,1] and T is called temperature and it decreases after each iteration following a prescribed schedule. The acceptance probability is high at the beginning and the probability of switching to neighbour less at the end of the algorithm. The possibility to select a

less optimal state *S'*, i.e., with higher $\delta(S')$ is meant to escape local minima where the algorithm could remain trapped.

**REF-C16:** P9, L9-21 - I have a few questions about this description. First, isn't the deviation of delta from 1 what is being optimized here? I don't see how this is explicit in the notation. The other question I have is about the size of the sub-ensemble. Can the procedure test sub-ensemble sizes all of the way to N-1 and all of the way down to some minimum number, maybe 2?

**Author-C16:** Yes, the deviation of delta from 1 is what is being optimized. To keep everything consistent we changed J by delta ($\delta$) as in equation 1 and following **REF-C15**. This change in symbol was applied to section 2.5 and 2.5.1.

Technically, it would be ideal to have the solutions for all ensemble sizes and evaluate which one is the minimum. Instead, we used an approach described in Garaud and Mallet (2011) to define the minimum size of the ensemble. To double-check their approach, we decided to test the method with three different ensemble sizes. We briefly explained how we select the size of the ensembles in section 2.5 and define the number of ensembles members in section 3.2. The paragraphs below from two different sections explain how we select the sub-ensembles size and establish that the calibration will be performed for three different sub-ensembles (ensemble size). We decided to add some lines to the document, where we specify that we can try all the potential solution, but for this study we decided to use a technique to decide that number of members:

Section 2.5, P8, L17-19:  In this study, we want to test the ability to reduce the ensemble from 45-members to an ensemble with smaller number of members that is still capable of representing the transport uncertainties and does not include members with redundant information. ***The number of ideal ensemble members could have been decided by performing the calibration for all the different size of ensemble smaller than 45-member. However, we decided to use an objective approach to select total number of members of the sub-ensemble. Therefore, we use the Garaud and Mallet (2011) technique to define the size of the calibrated sub-ensemble that each optimization technique will generate, the size of the sub-ensemble was determined by dividing the total number of observations by the maximum frequency in the large ensemble (45-members) rank histogram.*** We are going to generate sub-ensembles…"

**REF-C17:** P9, L30-31 - Is mutation a separate step here? Or is considered part of "crossover"?

**Auhtor-C17:**  In our genetic algorithm process, we only go through the selection and the crossover. We do not include a mutation process to the algorithm. Please find the edit version of these sentence below, to make the process clear.

P9, L30-31: "***Then this population will go through two out of the three steps of the genetic algorithm, (1) selection and (2) crossover.***"

**REF-C18:** P10, L103 - I have the same question that I had for the SA, are the sizes of sub-ensembles allowed to vary?

**Author-C18**: Yes, the size of the sub-ensembles can vary for Genetic Algorithm (see **Author-C16**).

**REF-C19:** P12, Section 3.2 - Does this answer my question about exploring the sizes of the sub-ensembles? One uses the largest frequency from the rank histogram and since this happens to be the first box, then than one gets used? Why are 5-member ensembles used?

**Author-C19:** Yes, this section as explained on **Author-C16** answers your question about the exploration different sub-ensemble sizes. To define this number, we used Garaud and Mallet (2011) technique as explained section 2.5, where the total number of observations is divided by maximum frequency of the full ensemble (45-members) histogram in our case the first bin of the histogram ($r_0$). Because we were not clear in the article about the rank histogram that was going to be used to define this number of members, we decided to add this to the following sentences:

P8, L17-19, Section 2.5: *"Therefore, we use the Garaud and Mallet (2011) technique to define the size of the calibrated sub-ensemble that each optimization technique will generate. The size of the sub-ensemble was determined by dividing the total number of observations by the maximum frequency in the large ensemble (45-members) rank histogram."*

P12, L19-20, Section 3.2: *"To compute the size of the sub-ensemble we use the maximum frequency of the rank histogram using the large ensemble (Figure 6). In this case the maximum frequency is the left bar ($r_0$) of every rank histogram."*

The maximum number of members that we could use based on Garaud and Mallet (2011) technique was 10 to 8 members based in the variable as explained in section 3.2. However, we decided to explore a smaller ensemble to see how this will change our results and also how this will end up contributing to future analysis such as the errors covariance.

**REF-C20:** P16, L7-9 - I'm struggling with the implication of this statement. It means that even though the sub-ensemble has the right spread it doesn't mean the simulation will encompass the true values? What about bias? If the model is biased one could get this situation, right?

**Author-C20:** The text was clarified. We are trying to explain here that the rank histogram score indicates that our calibrated ensembles have a good spread, but the spread-skill is telling us that our ensemble will not systematically encompass the true values for any given observation. Yes, our results show some biases in our model and therefore the ensemble. This bias can be associated to the model itself, the forcing data or the specific parameterization. We have minimized the bias, but future studies should perform model correction by using data assimilation or by improving the physics. We have modified the text to clarify our point.

Section 4.3, P16, L7-9: "The calibrated ensembles show the rank histogram score closer to one (Table 4), that is, flatter rank histograms (Figure 9) compared to the 45-member ensemble (Table 2 and Figure 6). The sub-ensembles do have a greater variance than the large ensemble (i.e., improved reliability) (Figure 14). However, the spread-skill relationship (i.e., resolution) of the calibrated ensembles do not show any major improvement compared to the 45-member ensemble, implying that the spread of the ensemble does not represent the day-to-day transport errors well. ***While the rank histogram suggests that the different calibrated ensembles have enough spread, the spread-skill relationship indicates that our ensemble does not systematically encompass the observations. The disagreement between the rank histogram and the spread-skill relationship can be associated with the metric used for the calibration (i.e., rank histogram) and the biases included in the calibrated ensemble. Using the score of the rank histogram alone may***

*not be sufficient to measure the reliability of the ensemble (Hamill, 2001), therefore, future down-selection studies should incorporate the resolution as part of the calibration process (skill score optimization). The biases in the model are a complex problem because there are many sources systematic errors within an atmospheric model (e.g., physical parameterizations, and meteorological forcing). Future studies should consider data assimilation or improvement of the physics parameterizations to reduce or remove these systematic errors. To improve the representation of daily model errors,* additional metrics should be introduced and the initial ensemble should offer a sufficient spread, possibly with additional physic parameterizations, additional random perturbations, or modifications of the error distribution of the ensemble (Roulston and Smith, 2003)."

**REF-C21:** P16, Section 4.4 - I'm not sure I follow this argument. I see from Fig 15 that the spatial correlations of CO2 get closer to 1 or -1, but I'm not sure why this happens with fewer ensemble members. It's stated that this is because of sample size (i.e. number of realizations?) but why should this result in a more intense correlation pattern? I would like to understand this.
**Author-C21:** This is an important and subtle point raised by the reviewer. We re-phrased the section 4.4 to clarify our conclusions. The concept of sampling noise can be compared to few random draws out of a complex distribution. It tends to generate spurious correlations (requiring a regularization similar to those used in the ensemble Kalman filters) depending on the shape of the true distribution. As a consequence, the error correlations increase or decrease randomly both near the observation location and at long distances. The variance of spurious correlations is in the order of 1/N with N the number of members (Bartlett, 1935, JRSS) and depends on the true distribution. In our case, considering the complexity of error distribution in space and time, we cannot predict the minimum size to avoid sampling noise but we clearly observe increased/decreased error correlations with 5 members.

*"Figure 15 shows the spatial correlation of 300 m DDA $CO_2$ errors with respect to the Round Lake site on DOY 180. Error correlations increase significantly as our ensemble size decreases. With fewer members, spurious correlations increase, resulting in high correlations at long distances. Assuming we sample only a few times the distribution of errors, our ensemble is very likely to be affected by spurious correlations with a variance on the order of 1/N."*